# Tetrahydroxanthohumol, a xanthohumol derivative, attenuates high-fat diet-induced hepatic steatosis by antagonizing PPARγ

Yang Zhang[1†]*, Gerd Bobe[2], Cristobal L Miranda[3], Malcolm B Lowry[4], Victor L Hsu[5], Christiane V Lohr[6], Carmen P Wong[1], Donald B Jump[1], Matthew M Robinson[7], Thomas J Sharpton[8], Claudia S Maier[9], Jan F Stevens[3], Adrian F Gombart[10†]*

[1]School of Biological and Population Health Sciences, Nutrition Program, Linus Pauling Institute, Oregon State University, Corvallis, United States; [2]Department of Animal Sciences, Linus Pauling Institute, Oregon State University, Corvallis, United States; [3]Department of Pharmaceutical Sciences, Linus Pauling Institute, Oregon State University, Corvallis, United States; [4]Department of Microbiology, Oregon State University, Corvallis, United States; [5]Department of Biochemistry and Biophysics, Oregon State University, Corvallis, United States; [6]Department of Biomedical Science, Carlson College of Veterinary Medicine, Corvallis, United States; [7]School of Biological and Population Health Sciences, Kinesiology Program, Oregon State University, Corvallis, United States; [8]Department of Microbiology, Department of Statistics, Oregon State University, Corvallis, United States; [9]Department of Chemistry, Linus Pauling Institute, Oregon State University, Corvallis, United States; [10]Linus Pauling Institute, Department of Biochemistry and Biophysics, Oregon State University, Corvallis, United States

*For correspondence:
zhangya3@oregonstate.edu (YZ);
adrian.gombart@oregonstate.edu
(AFG)

†These authors contributed
equally to this work

Competing interests: The
authors declare that no
competing interests exist.

Reviewing editor: Matthew A
Quinn, Wake Forest School of
Medicine, United States

**Abstract** We previously reported xanthohumol (XN), and its synthetic derivative tetrahydro-XN (TXN), attenuates high-fat diet (HFD)-induced obesity and metabolic syndrome in C57Bl/6J mice. The objective of the current study was to determine the effect of XN and TXN on lipid accumulation in the liver. Non-supplemented mice were unable to adapt their caloric intake to 60% HFD, resulting in obesity and hepatic steatosis; however, TXN reduced weight gain and decreased hepatic steatosis. Liver transcriptomics indicated that TXN might antagonize lipogenic PPARγ actions in vivo. XN and TXN inhibited rosiglitazone-induced 3T3-L1 cell differentiation concomitant with decreased expression of lipogenesis-related genes. A peroxisome proliferator activated receptor gamma (PPARγ) competitive binding assay showed that XN and TXN bind to PPARγ with an $IC_{50}$ similar to pioglitazone and 8–10 times stronger than oleate. Molecular docking simulations demonstrated that XN and TXN bind in the PPARγ ligand-binding domain pocket. Our findings are consistent with XN and TXN acting as antagonists of PPARγ.

## Introduction

Non-alcoholic fatty liver disease (NAFLD) is a major global health threat characterized by excessive hepatic lipid droplet accumulation with a history of little or no alcohol consumption (*Hashimoto et al., 2013*). About one-quarter of the US population suffers from NAFLD (*Estes et al., 2018*), with rates in the rest of the world ranging from 14% in Africa to 32% in the Middle East

(*Younossi et al., 2016*). The continuing obesity and diabetes epidemic drives increasing rates of NAFLD (*Estes et al., 2018*). Unfortunately, no FDA-approved drugs exist for its treatment. Sustained healthy lifestyle changes and weight loss are the only interventions proven effective in preventing the onset and progression of NAFLD (*Stefan et al., 2019*). Thus, there is a critical need for novel and effective interventions.

As a central hub for lipid metabolism, a healthy liver maintains homeostasis among uptake, esterification, oxidation, and secretion of fatty acids (FAs) (*Goldberg and Ginsberg, 2006*). Overconsumption of saturated FAs or sugars can overload the liver and disrupt lipid homeostasis, resulting in excess storage of triacylglycerols (TAG) in hepatocytes and the onset and progression of hepatic steatosis (*Ipsen et al., 2018*). Given that peroxisome proliferator activated receptor gamma (PPARγ) is important in hepatic lipogenesis (*Sharma and Staels, 2007*), it has attracted considerable attention as a therapeutic target for NAFLD (*Almeda-Valdés et al., 2009*).

Attenuated PPARγ activity in heterozygous *Pparg*-deficient (*Pparg*[+/−]) C57Bl/6J mice protects against high-fat diet (HFD)-induced obesity, liver steatosis, and adipocyte hypertrophy; however, treatment with the PPARγ agonist pioglitazone (PGZ) abrogates the protection against adipocyte hypertrophy (enlarged adipocytes) and decreases insulin sensitivity (*Kubota et al., 1999*), suggesting a potential beneficial use for PPARγ antagonists to treat hepatic steatosis. PPARγ antagonists tanshinone IIA (*Gong et al., 2009*), β-cryptoxanthine (*Goto et al., 2013*), protopanaxatriol (*Zhang et al., 2014*), isorhamnetin (*Zhang et al., 2016*), and Gleevec (*Choi et al., 2016*) improved multiple metabolic parameters in diet-induced obese (DIO) mice. These observations strongly suggest that moderate inhibition of PPARγ activity may reduce the risk for developing hepatic steatosis induced by diet, and PPARγ antagonists may be useful for the treatment and prevention of NAFLD.

Xanthohumol (XN), a prenylated flavonoid found in hops (*Humulus lupulus* L.), improves multiple parameters of MetS in rat and mouse models (*Legette et al., 2014*; *Miranda et al., 2016*; *Miranda et al., 2018*). Tetrahydroxanthohumol (TXN), a non-estrogenic synthetic XN derivative (*Figure 1*), appears more effective in ameliorating MetS in DIO mice than XN possibly due to its 5-, 10-, and 12-fold higher levels in the muscle, plasma, and liver, respectively, as compared with XN (*Miranda et al., 2018*). Both compounds likely mediate their benefits via multiple mechanisms. XN inhibits differentiation of preadipocytes and induces apoptosis in mature adipocytes (*Yang et al., 2007*; *Rayalam et al., 2009*), attenuates the function of SREBP-1 by repressing its maturation (*Miyata et al., 2015*) and induces beiging of white adipose tissue (WAT), decreases adipogenesis,

**Figure 1.** Structures of XN and its synthetic derivative, TXN.

and induces lipolysis (*Samuels et al., 2018*). We recently showed that XN and TXN significantly change gut microbiota diversity and abundance, alter bile acid metabolism, and reduce inflammation in mice fed a HFD (*Zhang et al., 2020*). Collectively, these data suggest both XN and TXN are effective for treatment of metabolic disorders and are promising candidates for NAFLD prevention and treatment.

In the present study, we show a daily oral intake of 0.035% TXN or 0.07% XN strongly suppresses diet-induced liver steatosis in C57Bl/6J male mice. Supervised machine learning of liver RNA-seq data identified perturbations in PPARγ signaling. Based on cell culture experiments, a PPARγ competitive binding assay and molecular docking studies, we provide evidence that XN and TXN act as novel PPARγ antagonists with moderate binding activity. Collectively, our findings suggest that appropriate functional antagonism of PPARγ is a logical approach to prevent and treat diet-induced liver steatosis and other related metabolic disorders. The structures of XN and TXN could serve as scaffolds for synthesis of more effective compounds to treat NAFLD.

## Results

### TXN attenuates HFD-induced weight gain and improves glucose homeostasis independent of caloric intake

As expected, C57Bl/6J mice on a 60% HFD (*Figure 2A*, solid blue line) gained more BW than mice on the low-fat diet (LFD) (*Figure 2A*, dotted black line) throughout the experimental period (week 1: $p < 0.05$; week 2–16: $p < 0.001$; repeated measures). TXN supplementation (*Figure 2A*, solid dark green line) attenuated HFD-induced BW gain throughout the experimental period (week 1: $p < 0.05$; week 2–16: $p < 0.001$; repeated measures). XN supplementation showed a dose-response effect: the higher dosage (HXN; *Figure 2A*, solid red line), but not the lower dosage (LXN; *Figure 2A*, solid yellow line), attenuated HFD-induced BW gain between weeks 8 and 16. When BW gain was expressed as % of initial BW, HFD-fed mice almost doubled their initial BW (+98.3 ± 2.7%), whereas TXN-treated mice gained 33% less (+66.2 ± 5.8%, $p < 0.0001$), and LFD-fed mice gained 53% less (+45.8 ± 4.3%, $p < 0.0001$) than HFD-fed mice (*Figure 2B*). Although not statistically significant, both LXN- and HXN-treated mice gained 7.5% and 11% less, respectively (90.0 ± 3.3%, $p = 0.20$; 87.6 ± 3.9%, $p = 0.07$; *Figure 2B*). In male C57Bl/6J mice, a BW of approximately 40 g is a critical tipping point from which metabolic dysfunction occurs (*van Beek et al., 2015*). After 16 weeks, mean BW for these mice was LFD (37.5 ± 1.1 g), HFD (50.3 ± 0.6 g), LXN (49.9 ± 1.1 g), HXN (47.4 ± 1.1 g), and TXN (42.2 ± 1.6 g).

Overtime, mice adapted to the HFD by consuming less food than LFD-fed mice (*Figure 2C*). However, the discrepancy in food consumption was insufficient to counteract the elevated caloric intake (*Figure 2D*). HXN-treated mice adapted better to the HFD, indicated by decreased food intake at weeks 1, 6–10, 13, and 16 ($p < 0.05$), and caloric intake ($p = 0.01$) compared to HFD control mice (*Figure 2C,D*), resulting in less BW gain. In contrast, the attenuated BW gain in TXN-treated mice was not accompanied by a significant reduction in food or caloric intake (*Figure 2C,D*).

To measure the effect of XN and derivatives on glucose homeostasis, we performed glucose tolerance test (GTT) after feeding the corresponding diets for 9 weeks. GTT results showed impaired glucose clearance in HFD control mice (*Figure 2—figure supplement 1A*, dashed blue line; *Figure 2—figure supplement 1B*). Compared to HFD control mice, TXN-treated mice showed significantly improved glucose clearance, as indicated at time points 30 min, 60 min, and 120 min post i.p. injection (*Figure 2—figure supplement 1A*, green line; p-values=0.04, 0.02, and <0.01, respectively), as well as a significant lower AUC (*Figure 2—figure supplement 1B*, $p < 0.01$). HXN-treated mice also showed improved glucose clearance at time points 60 min and 120 min post i.p. injection (*Figure 2—figure supplement 1A*, red line; p-values=0.04 and 0.05, respectively). Although not statistically significant, HXN-treated mice showed a trend toward a lower AUC (*Figure 2—figure supplement 1B*, $p = 0.067$). LXN treatment did not improve glucose clearance (*Figure 2—figure supplement 1A*, orange line; *Figure 2—figure supplement 1B*).

While fasting glycemia was not different between TXN-treated and HFD control mice after 16 weeks of feeding (*Figure 2—figure supplement 1C*, $p = 0.56$), fasting insulin was significantly improved by TXN treatment as suggested by lower circulating insulin (*Figure 2—figure supplement 1D*, $p = 0.003$) and HOMA-IR (*Figure 2—figure supplement 1E*, $p = 0.001$). These results indicate that

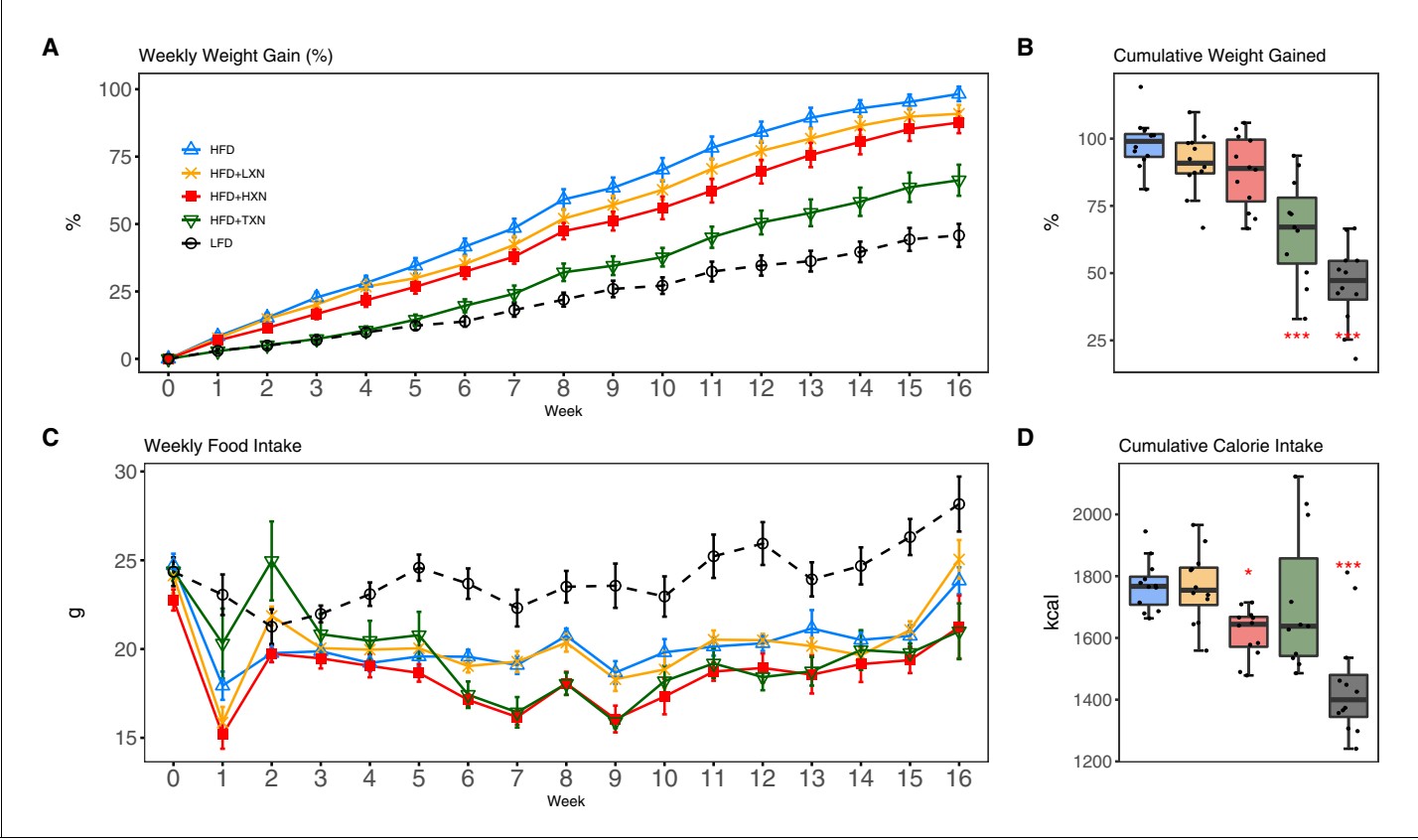

**Figure 2.** TXN and HXN suppress HFD-induced BW gain independent of caloric intake. Mice were fed either a LFD (black dashed line with empty circles, n = 12), a HFD (blue solid line with empty triangles, n = 12), HFD+LXN (yellow solid line with crosses, n = 12), HFD+HXN (red solid line with squares, n = 12), or HFD+TXN (green solid line with empty triangles, n = 11) for 16 weeks. (**A**) BW gain was assessed once per week. Data is expressed as means ± SEM. Repeated measurement of ANOVA was used to calculate p-values for the percentage of weight gained weekly. (**B**) Total percent BW gained at the end of the 16-week feeding period. Data is expressed as quartiles. (**C**) Food intake was assessed once per week during the 16 week feeding period. Data is expressed as means ± SEM. Repeated measurement of ANOVA was used to calculate p-values for weekly food intake. (**D**) Total calories consumed at the end of 16 week f eeding period. Data are expressed as quartiles. Source files of data used for the analysis and visualization are available in the *Figure 2—source data 1*.

The online version of this article includes the following source data and figure supplement(s) for figure 2:

**Source data 1.** Source files.
**Figure supplement 1.** TXN supplementation significantly improves glucose homeostasis in HFD-induced obese mice.

TXN significantly improved glucose homeostasis; XN seems to have a dose response as HXN appears to be more effective than LXN.

## TXN attenuates hepatic steatosis and HFD-induced obesity

HFD-induced BW gain was primarily body fat accumulation, as indicated by measurements obtained from DEXA scans. HFD mice had greater fat mass than LFD mice (p<0.0001; *Figure 3A*). Linear regression of total fat mass to total caloric intake revealed a strong relationship between caloric intake and fat mass among groups (r = +0.52; p<0.0001) and within LFD-fed mice (r = +0.79; p=0.002; *Figure 3A1*). In contrast, caloric intake was not correlated to fat mass in any HFD group (*Figures 3A*, 2-5), indicating a disconnection between caloric intake and fat mass after prolonged HFD consumption. Supplementation with HXN (−9.93%; p<0.05) and even more so with TXN (−27.7%; p<0.001) decreased body fat mass on HFD (*Figure 3A*), indicating that HXN and TXN attenuated the HFD-induced body fat accumulation and that this effect was not explained by changes to caloric intake (*Figures 3A*, 4-5).

Hepatic steatosis was measured by percent surface area occupied by lipid vacuoles in formalin-fixed, paraffin-embedded liver by image analysis of photomicrographs. In the absence of

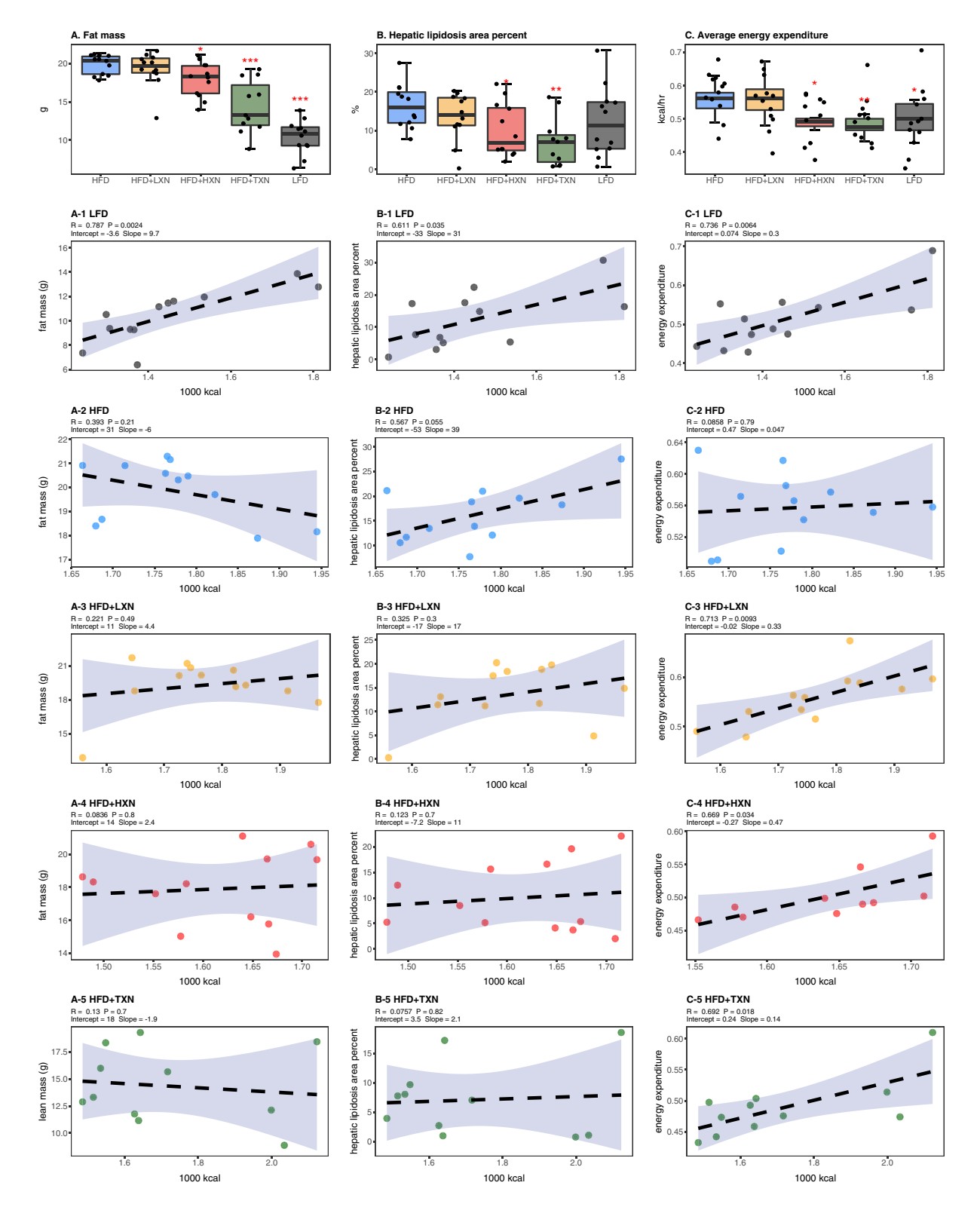

**Figure 3.** Energy homeostasis imbalance induced by HFD is prevented by XN and TXN supplementation. Mice were fed either a LFD (black, n = 12), a HFD (blue, n = 12), HFD+LXN (yellow, n = 12), HFD+HXN (red, n = 12), or HFD+TXN (green, n = 11) for 16 weeks. (A) Total fat mass measured by DXA scan 2 days prior to necropsy is expressed as quartiles. (A-1) Relationship between total fat mass and total caloric intake over 16 weeks of feeding for LFD; (A-2) HFD; (A-3) HFD+LXN; (A-4) HFD+HXN; and (A-5) HFD+TXN groups. (B) Hepatic lipidosis area percent expressed as quartiles. (B-1)
*Figure 3 continued on next page*

*Figure 3 continued*

Relationship between hepatic lipidosis area percent and total caloric intake over 16 weeks of feeding for LFD; (B-2) HFD; (B-3) HFD+LXN; (B-4) HFD+HXN; and (B-5) HFD+TXN groups. (C) Average energy expenditure over two light–dark cycles (48 hr) obtained using metabolic cages and expressed as quartiles. (C-1) Relationship between energy expenditure and total caloric intake over 16 weeks of feeding for LFD; (C-2) HFD; (C-3) HFD+LXN; (C-4) HFD+HXN (with removal of two outliers); (C-5) for HFD+TXN groups. Pre-planned general linear model with contrasts were used to calculate *p*-values in (A), (B), and (C). *p<0.05, **p<0.01, ***p<0.001. Linear regression analyses of total calories versus total fat mass (A1-5), hepatic lipidosis area percent (B1-5), and average energy expenditure (C1-5) in mice were done using stats package version 3.6.2 in R. Blue shading represents 95% CI of the regression line. Absolute values of R, p-value, intercept, and slope for the regression are reported above each corresponding panel. Source files of data used for the analysis are available in the *Figure 3—source data 1*.

The online version of this article includes the following source data and figure supplement(s) for figure 3:

**Source data 1.** Source files.
**Figure supplement 1.** Relationship of body mass and energy expenditure between (A) LFD and HFD; (B) LXN and HFD; (C) HXN and HFD; (D) TXN and HFD.
**Figure supplement 2.** The effect of diet and intervention on fasting plasma and fecal TAG levels.

supplementation, HFD- and LFD-fed mice shared similar hepatic lipid areas (*Figure 3B*). Caloric intake was positively correlated with hepatic lipid area on both LFD-fed mice (r = +0.61, p=0.03; *Figure 3B1*) and HFD-fed mice (r = +0.57, p=0.05; *Figure 3B2*). Supplementation with HXN (p<0.05) and TXN (p<0.01) mitigated hepatic steatosis, independent of caloric intake (*Figures 3B*, 4–5).

Changes in energy balance may drive changes in obesity-related steatosis. We investigated TXN on whole-body energy metabolism to determine mechanisms of TXN protection from weight gain, which can influence steatosis. Towards the end of the study, we measured whole-body expenditure for all 59 mice using a computer-controlled indirect calorimetry system (metabolic cages). Energy expenditure was calculated from the oxygen and carbon dioxide exchange ratio using the Weir equation (*Weir, 1949*). Total energy expenditure contains energy expenditure for basal metabolism, body tissue synthesis, digestion, and physical activity (*Speakman, 2013*). Mice consuming HFD and mice supplemented with LXN had higher (p<0.05) energy expenditure than mice on LFD, HXN, and TXN (*Figure 3C*). Caloric intake was positively correlated with energy expenditure in LFD- (*Figure 3C1*), LXN- (*Figure 3C3*), HXN- (*Figure 3C4*), and TXN-fed mice (*Figure 3C5*) but was not correlated with energy expenditure in HFD mice (*Figure 2C2*). We investigated the influence of body mass on energy expenditure using analysis of covariance (ANCOVA) of body mass upon entry into the cages between diets (*Tschöp et al., 2011*). ANCOVA revealed that LXN, HXN, or TXN supplementation did not change the positive relationship between energy expenditure and body mass (*Figure 3—figure supplement 1*).

As a marker of hepatic lipid uptake and export, fasting plasma TAG level was measured at the end of the study. Similar to hepatic lipid area, fasting plasma TAG did not reflect the caloric density of the diet (*Figure 3—figure supplement 2*); namely, there was an inverse relationship between caloric intake and plasma TAG among LFD mice (Spearman, r = −0.60, p=0.04; *Figure 3—figure supplement 2 A1*), which was lost on the HFD (Spearman, r = 0.12, p=0.70; *Figure 3—figure supplement 2 A2*). TXN treatment restored the negative correlation between caloric intake and plasma TAG (Spearman r = −0.65, p=0.04; *Figure 3—figure supplement 2 A5*). One explanation for the higher plasma TAG (p<0.01) observed could be that TXN inhibited hepatic lipid uptake, promoted hepatic lipid export, or both. TAG levels remained in the normal physiological range (40–60 mg/dl) for all groups (*Bogue et al., 2020*).

We collected fecal pellets over a 3 day period and measured fecal TAG at the end of the study as an indicator of fecal energy excretion. Fecal TAG levels did not differ among all groups (*Figure 3—figure supplement 2B*). No relationship was observed between caloric intake and fecal TAG among or within groups (*Figure 3—figure supplement 2 B1–5*), suggesting that the attenuated BW gain and hepatic steatosis in TXN- and HXN-treated mice was not related to increased fecal TAG excretion.

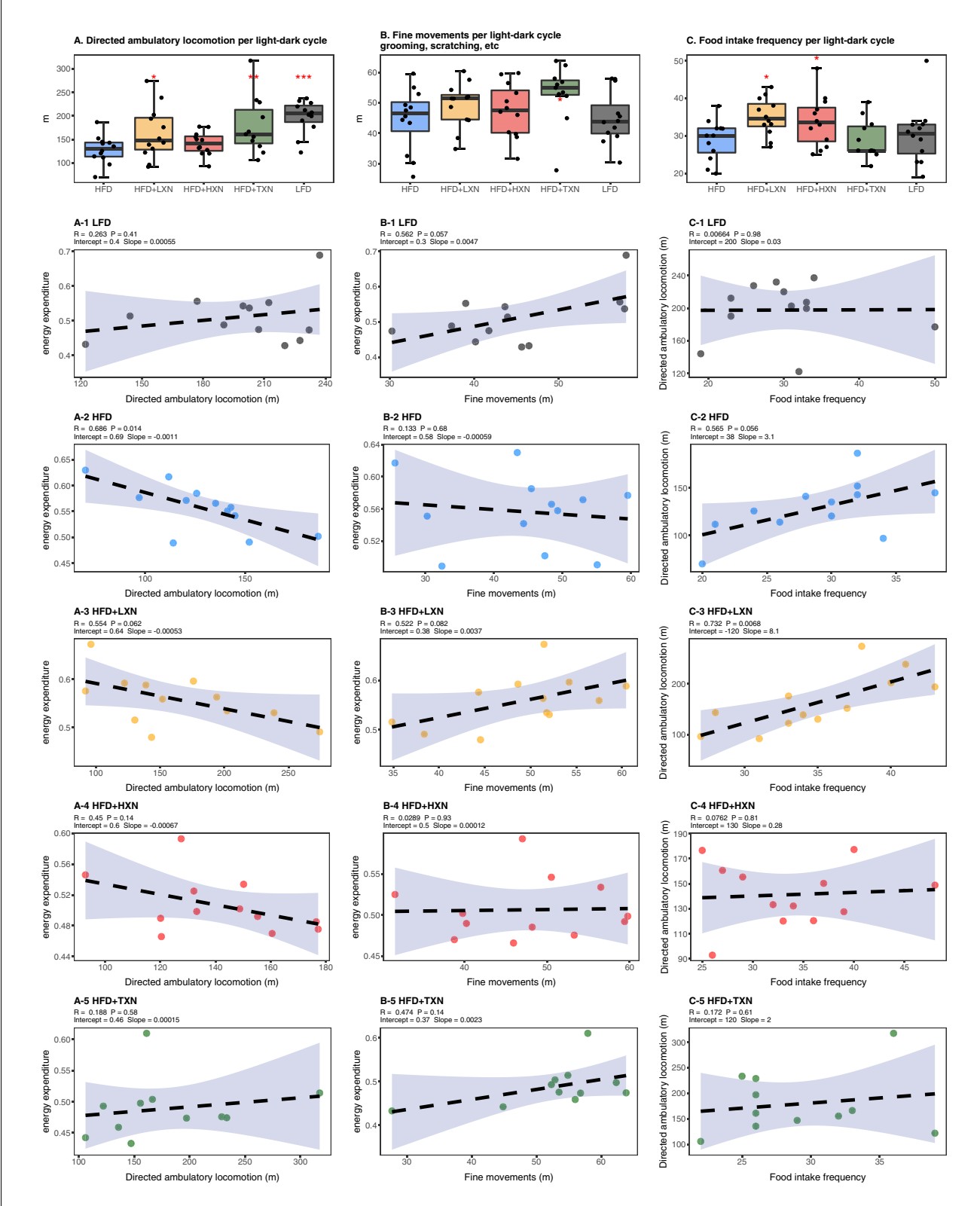

**Figure 4.** Effects of XN and TXN on food intake frequency, physical activity, and energy expenditure. Mice were fed either a LFD (black, n = 12), a HFD (blue, n = 12), HFD+LXN (yellow, n = 12), HFD+HXN (red, n = 12), or HFD+TXN (green, n = 11) for 16 weeks. (A) Directed ambulatory locomotion per 24 hr cycle obtained using a computer-controlled indirect calorimetry system. Data expressed as quartiles. (A-1) Relationship between directed ambulatory locomotion and energy expenditure for LFD; (A-2) HFD; (A-3) HFD+LXN; (A-4) HFD+HXN, and (A-5) HFD+TXN groups. (B) Fine movements

*Figure 4 continued on next page*

**Figure 4 continued**

per 24 hr cycle calculated by subtracting directed ambulatory locomotion from sum of all distances traveled within the beam-break system. Data is expressed as quartiles. (B-1) Relationship between fine movements and energy expenditure for LFD; (B-2) HFD; (B-3) HFD+LXN; (B-4) HFD+HXN; and (B-5) HFD+TXN groups. (C) Number of food intake events recorded in metabolic cages. Data expressed as quartiles. (C-1) Relationship between number of food intake events and directed ambulatory locomotion for LFD; (C-2) HFD; (C-3) HFD+LXN; (C-4) HFD+HXN; and (C-5) for HFD+TXN groups. Pre-planned general linear model with contrasts were used to calculate p-values in (A), (B), and (C). *p<0.05, **p<0.01, ***p<0.001. Linear regression analyses of energy expenditure versus directed ambulatory locomotion (A1-5), fine movements (B1-5), and number of food intake events (C1-5) in mice were done using stats package version 3.6.2 in R. Blue shading represents 95% CI of the regression line. Absolute values of R, p-value, intercept, and slope for the regression are reported above each corresponding panel. Source files of data used for the analysis are available in the *Figure 4—source data 1*.

The online version of this article includes the following source data for figure 4:

**Source data 1.** Source files.

## Effects of XN and TXN on food intake frequency, physical activity, and energy expenditure

We considered if physical activity level could explain attenuated weight gain of XN- and TXN-treated groups. We differentiated activity measured in the metabolic cages into directed ambulatory locomotion (sum of all locomotion of 1 cm/s or above within the x, y beam-break system) (*Figure 4A*) and fine movements (e.g., grooming, nesting, and scratching) (*Figure 4B*). In addition, we approximated the ambulatory movement for food consumption by measuring feeding frequency (*Figure 4C*). In contrast to energy expenditure (*Figure 3C*), directed ambulatory locomotion was lower in HFD- than LFD-fed mice (*Figure 4A*), while fine movement level (*Figure 4B*) and feeding frequency (*Figure 4C*) were not changed. TXN-treated mice exhibited higher directed ambulatory locomotion and fine movement levels than HFD mice (*Figure 4A,B*), whereas feeding frequency was unchanged (*Figure 4C*). XN-treated HFD mice showed higher directed ambulatory locomotion activity and feeding frequency than HFD mice (*Figure 4A,C*), whereas fine movement activity levels were not affected (*Figure 4B*).

In HFD-fed and LXN-treated mice, directed ambulatory locomotion levels were positively correlated with food frequency (*Figure 4C2-3*) but negatively correlated with energy expenditure (*Figure 4A2-3*), suggesting that food-driven activity may account for a major part of total directed ambulatory motion and that these mice spent the majority of their time and energy moving around for food consumption.

## TXN attenuates HFD-induced lipid accumulation in WAT

To assess the effect of XN and TXN on lipid accumulation, fat pads from three distinct sites –subcutaneous (sWAT), epididymal (eWAT), and mesenteric (mWAT) adipose tissue – were carefully removed and weighed during necropsy. Diet-induced lipid accumulation differed by adipose site. Compared to the LFD, the HFD-induced increase in mWAT fat mass was much greater than the increase in sWAT fat mass (3-fold vs. 2.5-fold increase, respectively), with the smallest increase (15%) observed in eWAT fat mass (*Figure 5A–C*). Supplementation with HXN (p<0.05), and even more so TXN (p<0.0001), decreased sWAT and mWAT fat mass. Compared to the HFD group, a smaller but significant increase in eWAT adipose tissue weight was observed in HXN-treated mice, while that of TXN-treated mice trended higher (p=0.06) (*Figure 5B*).

Caloric intake across diets was positively correlated with sWAT (r = +0.47; p=0.0002; *Figure 5A*) and mWAT fat mass (r = +0.39; p=0.002; *Figure 5C*), but no relationship was observed within XN- or TXN-treated groups (*Figure 5A3–5, C3–5*), indicating lipid accumulation in sWAT and mWAT fat depots was primarily linked to diet rather than the amount of food consumed. In eWAT adipose depot, we observed the opposite. Unlike sWAT and mWAT fat depots, caloric intake across diets was not correlated with eWAT fat mass (r = +0.03; p=0.82; *Figure 5B*). Instead, a positive correlation between caloric intake and eWAT fat mass was found within LFD-fed mice (*Figure 5B1*), and a negative correlation between caloric intake and eWAT fat mass was observed in both XN- and TXN-treated mice (*Figure 5B3–5*). No correlation was found in HFD-fed control mice (*Figure 5B2*). These observations are consistent with distinct WAT depots in mice differing in expandability (*van Beek et al., 2015*).

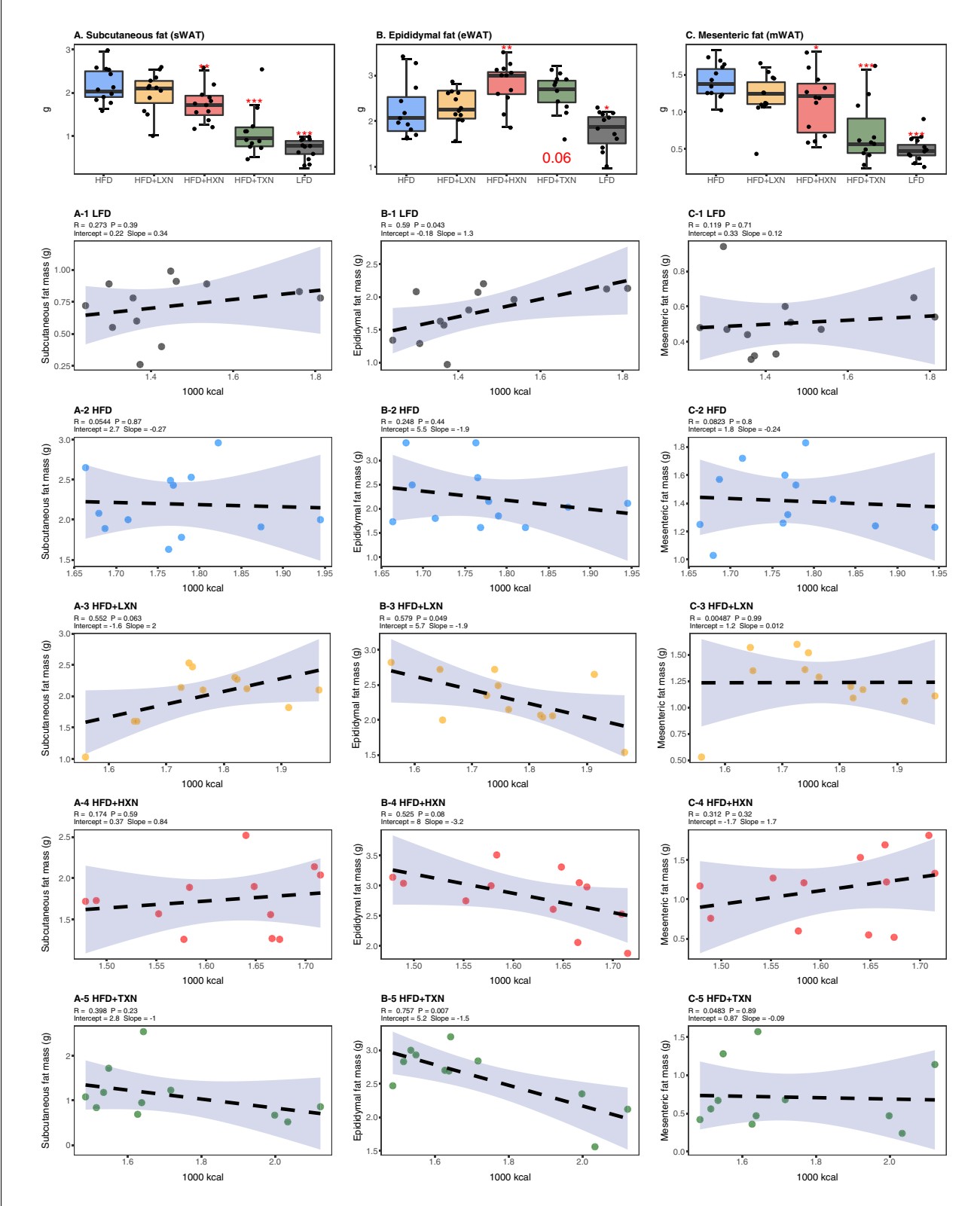

**Figure 5.** TXN decreases and alters the regional distribution of fat tissue accumulation. Mice were fed either a LFD (black, n = 12), a HFD (blue, n = 12), HFD+LXN (yellow, n = 12), HFD+HXN (red, n = 12), or HFD+TXN (green, n = 11) for 16 weeks. All fat masses were weighed on day of necropsy. (**A**) sWAT fat mass expressed as quartiles. (**A-1**) Relationship between sWAT fat mass and total caloric intake over 16 weeks of feeding for LFD; (**A-2**) HFD; (**A-3**) HFD+LXN; (**A-4**) HFD+HXN; and (**A-5**) HFD+TXN groups. (**B**) eWAT fat mass expressed as quartiles. (**B-1**) Relationship between eWAT fat mass

*Figure 5 continued on next page*

*Figure 5 continued*

and total caloric intake over 16 weeks of feeding for LFD; (**B-2**) HFD; (**B-3**) HFD+LXN; (**B-4**) HFD+HXN; and (**B-5**) HFD+TXN groups. (**C**) mWAT fat mass expressed as quartiles. (**C-1**) Relationship between mWAT fat mass and total caloric intake over 16 weeks of feeding for LFD; (**C-2**) HFD; (**C-3**) HFD +LXN; (**C-4**) HFD+HXN (with removal of two outliers); and (**C-5**), and HFD+TXN groups. Pre-planned general linear model with contrasts were used to calculate p-values in (**A**), (**B**), and (**C**). *p<0.05, **p<0.01, ***p<0.001. Linear regression analyses of total calories versus sWAT (**A1-5**), eWAT (**B1-5**), and mWAT fat masses (**C1-5**) in mice were done using stats package version 3.6.2 in R. Blue shading represents 95% CI of the regression line. Absolute values of R, p-value, intercept, and slope for the regression are reported above each corresponding panel. Source files of data used for the analysis are available in *Figure 5—source data 1*.

The online version of this article includes the following source data for figure 5:

**Source data 1.** Source files.

## HXN and TXN protect against NAFLD on a HFD

NAFLD is characterized by accumulation of number and size of intrahepatic microvesicular and macrovesicular lipid vacuoles. Mice on a LFD diet possessed hepatic lipid vacuoles and resembled livers of low-density lipoprotein receptor knock-out (LDLR$^{-/-}$) mice on a similar synthetic diet (*Lytle and Jump, 2016*); however, their liver to BW ratio of about 4% was in a normal healthy range (*Lytle et al., 2017*). HFD-fed mice had many smaller lipid vacuoles (*Figure 6A*). XN supplementation decreased the number and size of intrahepatic lipid vacuoles in HFD mice in a dose-dependent manner (*Figure 6A*). Supplementation with TXN almost completely prevented hepatic lipid vacuole accumulation in HFD mice, resulting in less lipid accumulation than in LFD mice (*Figure 6A*). We did not detect discernable fibrosis in liver sections using Sirius red staining in any of the mice (data not shown).

The liver to BW ratio is an indicator of NAFLD with a ratio above 4% indicating NAFLD (*Lytle et al., 2017*). The majority of mice (10 of 12) on a HFD diet had a liver to BW ratio above 4.5%, whereas all LFD mice had a liver to BW ratio between 3.8% and 4.3% (*Figure 6B*). Supplementation with HXN decreased the number of mice with a liver to BW ratio above 4% to 4 of 12 mice and all TXN-supplemented mice had a liver to BW ratio below 3.6% except for one, which had a liver to BW ratio of 4% (*Figure 6B*). These data are consistent with TXN and, to a smaller extent, HXN reducing NAFLD. Hepatic lipid extracts from TXN-supplemented HFD mice and LFD-fed mice had lower liver triglyceride concentrations than from mice fed with HFD, LXN, or HXN (*Figure 6C*).

Another indicator of NAFLD is the liver area occupied by lipids; the histological lower cutoff for NAFLD is over 5% of liver area (*Brunt, 2010*). Using this cutoff, all control HFD mice had NAFLD and 10 of 12 LFD mice had NAFLD (*Figure 3B*). Both HXN and TXN supplementation decreased liver lipid accumulation on a HFD by twofold (*Figure 3B*). Three of 12 HXN-supplemented mice and 5 of 11 TXN-supplemented mice had less than 5% lipid area, while 7 of 12 HXN-supplemented mice and 9 of 11 TXN-supplemented mice had less than 10% lipid area (*Figure 3B*). In comparison, only 1 of 12 HFD control mice were below 10% lipid area. The supplement-induced decrease was independent of caloric intake (*Figure 3B3-5*).

## RNA-seq reveals suppression of hepatic FA biosynthesis processes and pathways by HXN and TXN treatments

We conducted RNA-seq analysis of the livers obtained from mice after 16 weeks on the diet to determine transcriptional mechanisms by which HXN and TXN supplementation could ameliorate hepatic steatosis induced by HFD. Gene counts were calculated to quantify gene expression in the four diet groups: LFD, HFD, HFD+HXN, and HFD+TXN. The differentially expressed genes (DEGs) were determined using a false discovery rate (FDR) cutoff of <0.4, as compared to HFD.

To visualize expression patterns of DEGs in the four groups, we used hierarchical clustering with a heat map (*Figure 7A*). The DEGs clustered into two major types, one with higher expression (red) in the LFD and HFD groups but lower expression (blue) in the HXN and TXN groups and the other with lower expression in the LFD and HFD groups but higher expression in the HXN and TXN groups (*Figure 7A*). Individual mice clustered into two major nodes. All HFD mice clustered with six LFD and four HXN mice and all TXN mice clustered with six HXN and four LFD mice (*Figure 7A*). This likely reflects the variability observed in phenotypic outcomes (*Figure 6B*). The volcano plot analysis of gene expression revealed that both HXN and TXN treatments induced significant changes in gene expression compared with the HFD group (*Figure 7B*). TXN treatment had the greatest effect

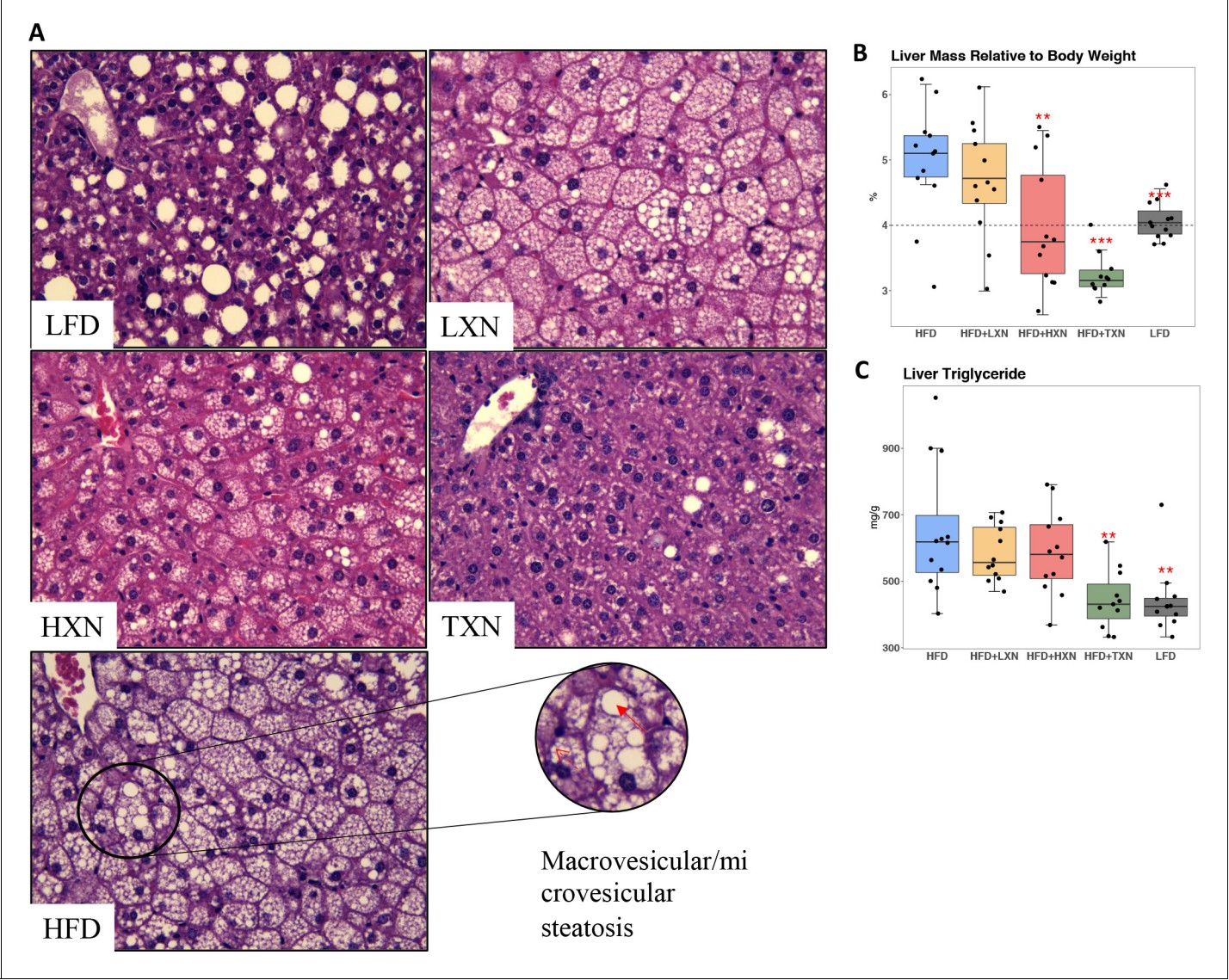

**Figure 6.** TXN prevents HFD-induced liver steatosis in mice. Mice were sacrificed at the end of the study and liver samples were freshly collected and processed. (**A**) Representative histological images of H and E staining of liver sections. An enlarged image representative of a liver section from a HFD-fed mouse is shown as a circle on the bottom right. Macrovesicular steatosis or large lipid droplets are indicated by the red bold arrow; microvesicular steatosis or small lipid droplets are indicated by the broken red line arrow. (**B**) Liver mass to BW ratio. (**C**) Hepatic triglyceride content. P-values of orthogonal a priori comparisons of the HFD versus each of the other groups are shown. **$p<0.01$, ***$p<0.001$. Source files of data used for the analysis are available in *Figure 6—source data 1* and *2*.

The online version of this article includes the following source data for figure 6:

**Source data 1.** Source files for histology data.
**Source data 2.** This zip archive contains the following.

with 295 identified DEGs, while HXN treatment only resulted in six DEGs. We identified 212 DEGs in comparing the LFD and HFD groups.

We next conducted gene ontology (GO) enrichment and pathway analysis of DEGs using Enrichr (*Chen et al., 2013*). We assigned the DEGs in the TXN treatment group to GO terms describing biological processes. The enriched GO terms and pathways with adjusted p-values<0.05 are summarized in *Figure 8*, *Figure 8—source data 1*. GO enrichment analysis indicated that TXN treatment significantly downregulated genes involved in biological processes including xenobiotic catabolism,

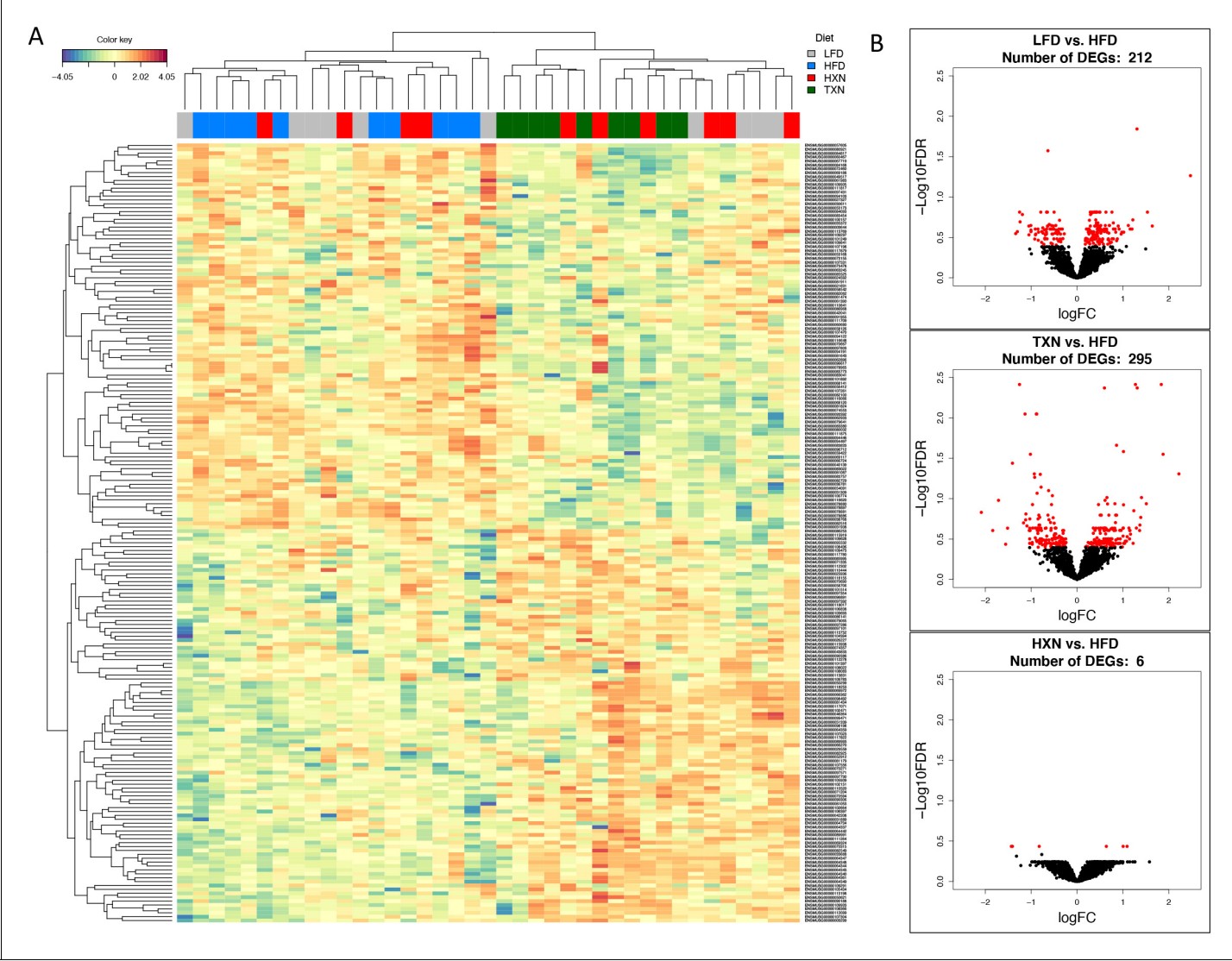

**Figure 7.** TXN treatment significantly alters liver transcriptome of mice after 16 weeks of feeding. (**A**) Hierarchical clustering of the top 200 differentially expressed genes (DEGs) in each treatment group (labeled at the top right corner: gray indicates LFD group, blue indicates HFD, red indicates HXN, and green indicates TXN.) as determined by RNA-seq analysis. Color key is based on the log$_2$ fold change. (**B**) Volcano plots show DEGs (red dots) in the comparison of different treatment groups. Source files of data used for the analysis are available in *Figure 7—source data 1*.

The online version of this article includes the following source data for figure 7:

**Source data 1.** Source files.

FA metabolism, glucose metabolism, and regulation of lipid metabolism (*Figure 8*, top panel). Furthermore, Kyoto Encyclopedia of Genes and Genomes (KEGG) pathway analysis demonstrated that TXN upregulated expression of genes in six pathways including complement and coagulation cascades, prion diseases, steroid hormone biosynthesis, arachidonic acid metabolism, retinol metabolism, and linoleic acid metabolism (*Figure 8*, bottom right panel). Many of these included genes encoding Cyp450 enzymes and genes from the major urinary protein family (*Table 1*). On the other hand, expression of genes in 25 KEGG pathways were significantly downregulated by TXN treatment compared to HFD (*Figure 8*, bottom left panel). The top 10 significantly enriched KEGG pathways based on statistical significance and combined score ranking included the biosynthesis of unsaturated FAs, glutathione metabolism, amino sugar and nucleotide sugar metabolism, glycolysis and gluconeogenesis, pentose phosphate pathway, fluid shear stress and atherosclerosis, chemical carcinogenesis, drug metabolism, FA elongation, and the PPAR signaling pathway. Consistent with the

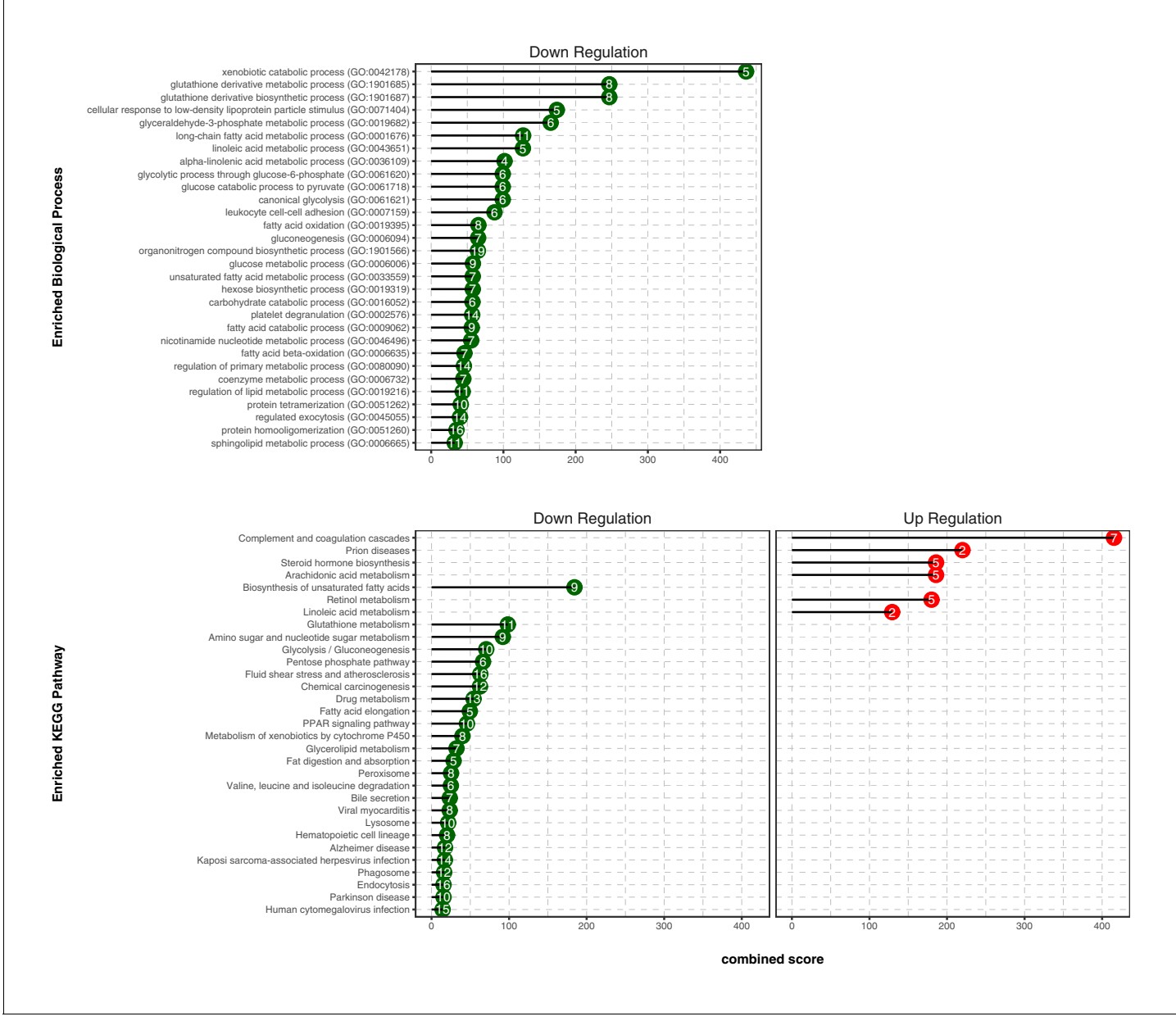

**Figure 8.** TXN decreases expression of numerous gene ontology and KEGG pathways. Analysis of DEGs from the livers of mice that consumed a HFD +TXN versus a HFD revealed mostly downregulation of biological processes and KEGG pathways. The significant (adjusted p<0.05) enriched biological process terms in gene ontology (upper panel) and enriched KEGG pathways (lower panel) were selected by Enrichr Tools based on significance and combined scores. The number inside each lollipop represents the number of identified DEG genes in that specific biological process or KEGG pathway. Source files of data used for the analysis are available in the *Figure 8—source data 1*.

The online version of this article includes the following source data for figure 8:

**Source data 1.** Source files.

lack of Sirius red staining in the liver, we observed no changes in expression of genes involved in hepatic fibrosis in the HFD mice compared with the LFD. In response to TXN treatment, we noted a fourfold decrease in *Timp2* and *Col1a1* both factors that promote hepatic fibrosis (*Table 2*) *Nie et al., 2004*; *Chakraborty et al., 2012*). We also did not observe changes in expression for transforming growth factor β1 (*Tgfb1*) or platelet-derived growth factor (*Pdgf*), key factors in driving hepatic stellate cell activation following hepatocellular injury (data not shown) (*Dooley et al., 2001*; *Tsuchida and Friedman, 2017*). Finally, we did not observe increased expression of genes involved

**Table 1.** Changes in transcript levels for genes encoding lipocalin two and hepatic major urinary proteins.

| Gene name | Gene symbol | HFD vs. LFD log2FC | FDR | TXN vs. HFD log2FC | FDR |
|---|---|---|---|---|---|
| Lipocalin 2 | Lcn2 | 0.60 | 0.66 | −1.60 | 0.08 |
| Major urinary protein 1 | Mup1 | −1.72 | 0.04 | 2.19 | <0.01 |
| Major urinary protein 2 | Mup2 | −0.77 | 0.29 | 1.38 | 0.01 |
| Major urinary protein 3 | Mup3 | −0.51 | 0.54 | 0.83 | 0.17 |
| Major urinary protein 4 | Mup4 | −1.24 | 0.03 | 1.22 | 0.02 |
| Major urinary protein 5 | Mup5 | −1.27 | 0.05 | 1.31 | 0.03 |
| Major urinary protein 6 | Mup6 | −0.94 | 0.12 | 1.13 | 0.03 |
| Major urinary protein 7 | Mup7 | −2.03 | 0.05 | 2.73 | <0.01 |
| Major urinary protein 8 | Mup8 | −1.75 | 0.03 | 2.16 | <0.01 |
| Major urinary protein 9 | Mup9 | −1.82 | 0.03 | 2.10 | <0.01 |
| Major urinary protein 10 | Mup10 | −0.70 | 0.31 | 1.28 | 0.01 |
| Major urinary protein 11 | Mup11 | −1.45 | 0.12 | 1.81 | 0.02 |
| Major urinary protein 12 | Mup12 | −2.21 | 0.05 | 2.65 | 0.01 |
| Major urinary protein 13 | Mup13 | −0.84 | 0.24 | 1.37 | 0.01 |
| Major urinary protein 14 | Mup14 | −0.89 | 0.22 | 1.47 | 0.01 |
| Major urinary protein 15 | Mup15 | −1.93 | 0.07 | 2.63 | <0.01 |
| Major urinary protein 16 | Mup16 | −1.17 | 0.13 | 1.40 | 0.04 |
| Major urinary protein 17 | Mup17 | −1.72 | 0.06 | 1.74 | 0.04 |
| Major urinary protein 18 | Mup18 | −0.97 | 0.29 | 1.27 | 0.08 |
| Major urinary protein 20 | Mup20 | −1.14 | <0.001 | 0.15 | 0.79 |
| Major urinary protein 21 | Mup21 | −0.97 | 0.07 | 1.15 | 0.02 |
| Major urinary protein 22 | Mup22 | −0.70 | 0.32 | 1.27 | 0.02 |

Genes with significant change after HFD feeding and with TXN treatment are highlighted in red (FDR ≤ 0.05).

in inflammation with 16 weeks of HFD feeding, but did observe a significant decrease in *Ccr2* and *Fgf21* expression with TXN treatment (*Table 3*).

We then examined transcript levels for genes in pathways regulated by PPARα, namely lipid oxidation. We observed no change with TXN treatment (*Table 4*). Consistent with the GO enrichment analysis, most of the changes were for genes encoding proteins involved in the lipid storage pathway (*Table 4*) and regulated by PPARγ.

## Identification of key hepatic genes regulated by TXN and involved in ameliorating hepatic steatosis

We implemented support vector machine (SVM) to identify a set of signature genes that can distinguish TXN-treated mice from HFD-fed control mice. Briefly, we used the DaMirSeq R package to determine a set of genes whose principal components best correlated with TXN treatment by performing backward variable elimination with partial least-squares regression and removing redundant features by eliminating those that were very highly correlated (*Chiesa et al., 2018*). Repeating this process 30 times, we used all 13 genes identified here as input into our SVM models (*Figure 9*, left panel). Genes identified classified HFD- and TXN-fed mice into two distinct groups (*Figure 9*, right panel). Eight of 13 genes showed significant, differential expression between TXN and HFD diet samples (*Table 5*). Consistent with the GO analysis, three of the eight genes – uncoupling protein 2 (*Ucp2*), cell death-inducing DFFA-like effector c (*Cidec*), and monoacylglycerol O-acyltransferase 1 (*Mogat1*) – are involved in lipid metabolism and are known target genes of PPARγ (*Medvedev et al., 2001*; *Kim et al., 2008*; *Matsusue et al., 2008*); (*Bugge et al., 2010*; *Karbowska and Kochan, 2012*; *Wolf Greenstein et al., 2017*).

We then confirmed expression of these genes using RT-qPCR. Consistent with RNA-seq results, TXN-treated mice had significantly lower expression of *Pparg2* and major PPARγ target genes *Cidec*,

**Table 2.** Changes in transcript levels for gene markers of hepatic fibrosis.

| Gene name | Gene symbol | HFD vs. LFD log2FC | FDR | TXN vs. HFD log2FC | FDR |
|---|---|---|---|---|---|
| Collagen, type 1, alpha 1 | Col1a1 | −0.01 | 1.00 | −1.92 | 0.09 |
| Collagen, type 1, alpha 2 | Col1a2 | 0.05 | 0.98 | −1.51 | 0.11 |
| Lysyl oxidase-like 1 | Loxl1 | −0.63 | 0.55 | −0.42 | 0.68 |
| Lysyl oxidase-like 2 | Loxl2 | 0.48 | 0.60 | −0.77 | 0.26 |
| Lysyl oxidase-like 3 | Loxl3 | −0.39 | 0.78 | −0.54 | 0.62 |
| Matrix metallopeptidase 12 | Mmp12 | 0.42 | 0.83 | −2.82 | 0.02 |
| Matrix metallopeptidase 14 | Mmp14 | −0.19 | 0.67 | −0.04 | 0.93 |
| Matrix metallopeptidase 15 | Mmp15 | −0.27 | 0.44 | 0.11 | 0.78 |
| Matrix metallopeptidase 19 | Mmp19 | 0.38 | 0.29 | −0.08 | 0.87 |
| Matrix metallopeptidase 2 | Mmp2 | −0.04 | 0.98 | −0.99 | 0.36 |
| Transforming growth factor alpha | Tgfa | 0.23 | 0.68 | 0.05 | 0.94 |
| Transforming growth factor beta 1 | Tgfb1 | 0.02 | 0.99 | 0.23 | 0.83 |
| Transforming growth factor beta 1 induced transcript 1 | Tgfb1i1 | 0.07 | 0.96 | −0.18 | 0.87 |
| Transforming growth factor beta 2 | Tgfb2 | −0.65 | 0.68 | −0.84 | 0.55 |
| Transforming growth factor beta 2 induced | Tgfbi | −0.08 | 0.92 | −0.53 | 0.26 |
| Transforming growth factor beta receptor I | Tgfbr1 | −0.24 | 0.70 | −0.20 | 0.73 |
| Transforming growth factor beta receptor II | Tgfbr2 | 0.14 | 0.85 | −0.68 | 0.15 |
| Transforming growth factor beta receptor III | Tgfbr3 | 0.17 | 0.79 | 0.004 | 1.00 |
| Tissue inhibitor of metalloproteinase 2 | Timp2 | −0.34 | 0.65 | −1.92 | 0.09 |
| Tissue inhibitor of metalloproteinase 3 | Timp3 | −0.48 | 0.31 | −1.51 | 0.11 |

Genes with significant change after HFD feeding and with TXN treatment are highlighted in red (FDR ≤ 0.05).

*Mogat1*, and *Plin4* (*Dalen et al., 2004*; *Fang et al., 2016*; *Figure 10*, top panel). Moreover, we observed significantly strong positive correlations between the expression of these three genes (*Figure 10*, bottom panel). The above results suggest TXN treatment inhibits the PPARγ pathway – a key pathway involved in hepatic lipid metabolism.

## XN and TXN attenuate intracellular lipid content in 3T3-L1 adipocytes in a dose-dependent manner

We hypothesized that TXN and XN antagonizes the PPARγ receptor, which would explain the decreased expression of its target genes. To test our hypothesis, we utilized 3T3-L1 murine fibroblast cells, which depend on PPARγ activity to differentiate into adipocytes (*Tamori et al., 2002*). XN and its derivatives are cytotoxic to some cells and to ensure that we used concentrations that were not cytotoxic to 3T3-L1 adipocytes, we tested an escalating dose of XN and TXN (*Strathmann and Gerhauser, 2012*). After treatments, we determined the number of live cells using an MTT assay. XN and TXN were only significantly cytotoxic for 3T3-L1 cells at a dose of 50 μM (data not shown). While it is difficult to translate in vivo doses to in vitro doses, based on previous in

**Table 3.** Changes in transcript levels for gene markers of hepatic inflammation.

| Gene name | Gene symbol | HFD vs. LFD log2FC | FDR | TXN vs. HFD log2FC | FDR |
|---|---|---|---|---|---|
| Adhesion G protein-coupled receptor E1 | Adgre | 0.13 | 0.87 | −0.37 | 0.50 |
| Chemokine ligand 2 | Ccl2 | 0.87 | 0.50 | −1.40 | 0.16 |
| Chemokine receptor 2 | Ccr2 | 0.86 | 0.32 | −1.84 | <0.01 |
| Fibroblast growth factor 21 | Fgf21 | 1.00 | 0.34 | −1.73 | 0.04 |
| Prostaglandin-endoperoxide synthase 1 | Ptgs1 | −0.08 | 0.92 | −0.09 | 0.89 |

Genes with significant change after HFD feeding and with TXN treatment are highlighted in red (FDR ≤ 0.05).

**Table 4.** Changes in transcript levels for genes encoding proteins involved in hepatic lipid oxidation, VLDL export, and lipid storage pathways.

| Gene name | Gene symbol | HFD vs. LFD log2FC | FDR | TXN vs. HFD log2FC | FDR |
|---|---|---|---|---|---|
| Lipid oxidation | | | | | |
| Acyl-CoA thioesterase 1 | Acot1 | −0.80 | 0.02 | 0.05 | 0.93 |
| Acyl-CoA oxidase 1 | Acox1 | 0.28 | 0.31 | −0.13 | 0.66 |
| Acyl-CoA oxidase 2 | Acox2 | 0.14 | 0.63 | 0.18 | 0.42 |
| Acyl-CoA oxidase 3 | Acox3 | 0.08 | 0.87 | −0.09 | 0.84 |
| Carnitine palmitoyltransferase 1a | Cpt1a | −0.08 | 0.81 | −0.16 | 0.46 |
| Carnitine palmitoyltransferase 2 | Cpt2 | −0.01 | 0.98 | 0.19 | 0.51 |
| ELOVL family member 5, elongation of long-chain fatty acids | Elovl5 | 0.40 | 0.32 | −0.89 | <0.01 |
| Elongation of very long-chain fatty acids | Elovl2 | −0.34 | 0.34 | 0.02 | 0.98 |
| 3-Hydroxy-3-methylglutaryl-Coenzyme A synthase 2 | Hmgcs2 | 0.26 | 0.25 | −0.06 | 0.84 |
| Peroxisome proliferator activated receptor alpha | Ppara | −0.23 | 0.77 | 0.35 | 0.56 |
| Solute carrier family 25 member 20 | Slc25a20 | 0.08 | 0.80 | −0.14 | 0.56 |
| VLDL export | | | | | |
| Apolipoprotein B | Apob | −0.07 | 0.86 | 0.02 | 0.95 |
| Diacylglycerol O-acyltransferase 1 | Dgat1 | −0.01 | 0.98 | −0.09 | 0.83 |
| Microsomal triglyceride transfer protein | Mttp | −0.20 | 0.81 | 0.41 | 0.48 |
| Lipid storage | | | | | |
| Cell death-inducing DFFA-like effector c | Cidec | 1.09 | 0.30 | −2.41 | <0.01 |
| Monoacylglycerol O-acyltransferase 1 | Mogat1 | 1.71 | <0.01 | −1.62 | 0.01 |
| Perilipin 2 | Plin2 | −0.11 | 0.82 | −0.30 | 0.33 |
| perilipin 3 | Plin3 | 0.20 | 0.66 | −0.51 | 0.08 |
| Perilipin 4 | Plin4 | 0.86 | 0.13 | −1.11 | 0.02 |
| Perilipin 5 | Plin5 | −0.39 | 0.19 | 0.01 | 0.98 |
| Peroxisome proliferator activated receptor gamma | Pparg | 0.97 | 0.44 | −1.14 | 0.26 |
| Peroxisome proliferator activated receptor gamma coactivator 1 alpha | Ppargc1a | −0.01 | 0.99 | −0.18 | 0.62 |
| Peroxisome proliferator activated receptor gamma coactivator 1 beta | Ppargc1b | −0.32 | 0.42 | 0.07 | 0.88 |

Genes with significant change after HFD feeding and with TXN treatment are highlighted in red (FDR ≤ 0.05).

vitro studies (*Yang et al., 2007*; *Samuels et al., 2018*) and our current cell viability data, we selected low (5 µM), medium (10 µM), and high (25 µM) concentrations of XN and TXN for the subsequent experiments where cell viability was greater than 90% (data not shown).

Murine preadipocyte 3T3-L1 differentiation and adipogenesis was induced by the addition of dexamethasone, 3-isobutyl-1-methylxanthine (IBMX), and insulin, which strongly induced intracellular lipid accumulation (*Figure 11A2-3*). Addition of XN significantly attenuated intracellular lipid levels in a dose-dependent manner (*Figure 11B1-3*). Like XN, TXN also strongly inhibit intracellular lipid accumulation (*Figure 11C1-3*).

## XN and TXN inhibit RGZ-induced adipocyte differentiation in 3T3-L1 cells in a dose-dependent manner

RGZ is a known potent PPARγ agonist used as an insulin-sensitizing agent. To test the hypothesis that XN and TXN may antagonize a known PPARγ ligand, we determined if the compounds would block RGZ-induced PPARγ actions (*Figure 12*). 3T3-L1 cells were treated with 0.1% DMSO, 1 µM rosiglitazone (RGZ), 1 µM GW9662, XN (5, 10, and 25 µM), TXN (5, 10, and 25 µM), 25 µM XN + 1 µM RGZ, or 25 µM TXN + 1 µM RGZ for 48 hr. RGZ strongly induced the differentiation (*Figure 12A1*), and GW 9662, a potent PPARγ antagonist, inhibited the RGZ-induced differentiation (*Figure 12A2*). We also observed that both XN (*Figure 12B1-3*) and TXN (*Figure 11C1-3*)

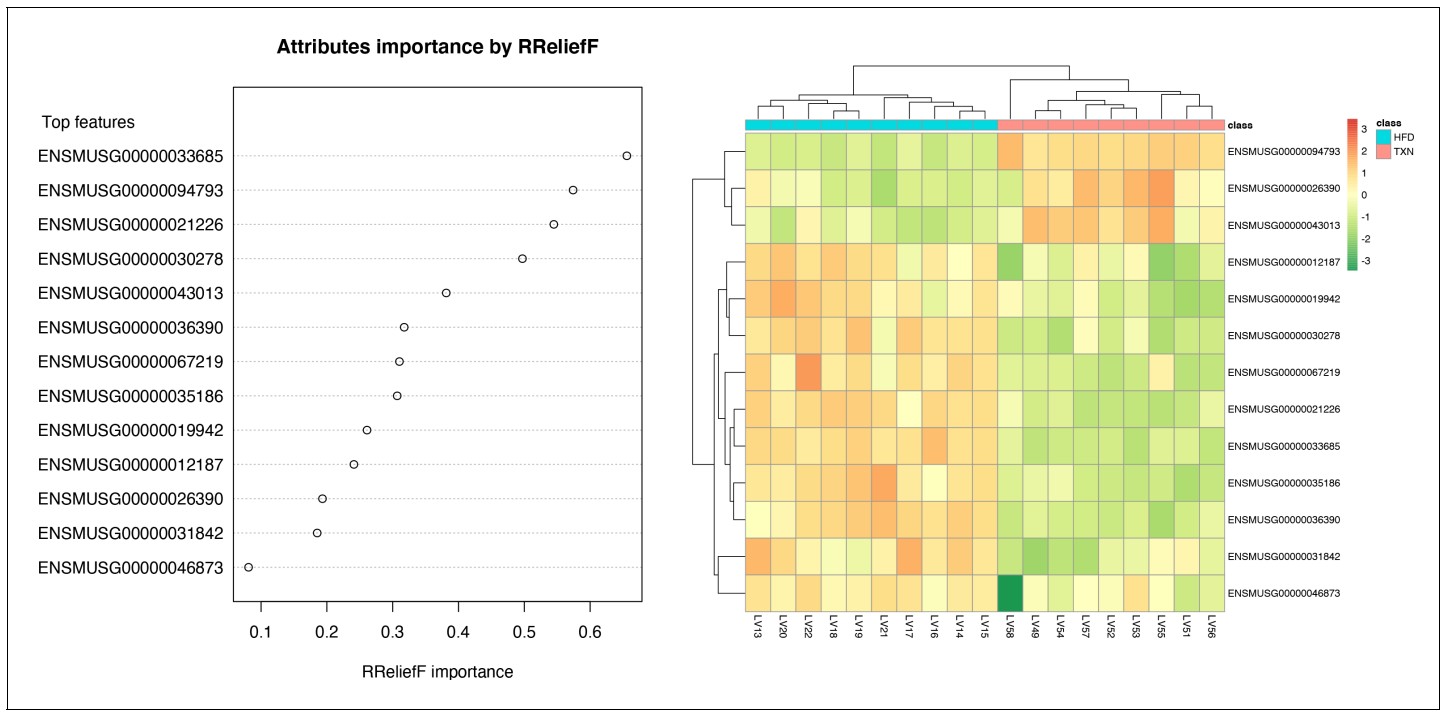

**Figure 9.** SVM identified signature genes that distinguish mice that consumed TXN. Left panel: The dot chart shows the top 13 genes, sorted by RReliefF importance score. This plot was used to select the most important predictors to be used for classification. Right panel: Colors in the heatmap highlight the gene expression level in fold change: color gradient ranges from *dark orange*, meaning 'upregulated', to *dark green*, meaning 'downregulated'. On the top of the heatmap, horizontal bars indicate HFD (blue) and HFD+TXN (pink) treatments. On the top and on the left side of the heatmap, the dendrograms obtained by Spearman's correlation metric are shown. Plots were produced with DaMiRseq R package 1.10.0. Source files of data used for the analysis are available in *Figure 9—source data 1*.

The online version of this article includes the following source data for figure 9:

**Source data 1.** Source files.

## XN and TXN downregulate genes regulated by PPARγ in 3T3-L1 cells

To elucidate the effect of XN and TXN on PPARγ action at the transcriptional level, we measured the expression of several known PPARγ target genes using RT-qPCR on samples 7 days post 25 µM XN or TXN treatment. Consistent with the decrease of intracellular lipid content in *Figures 11* and *12*, the expression of *Pparg* and its target genes at 7 days post-treatment were significantly downregulated by XN and TXN treatments (*Table 6*). Cells treated with 1 µM GW 9662, a PPARγ antagonist, did not significantly reverse the RGZ-induced upregulation of these genes. Cells treated with either 25 µM XN or TXN significantly reversed the RGZ-induced upregulation of *Cd36* (p<0.001, p<0.001), *Fabp4* (p<0.001, p<0.001), *Mogat1* (p<0.001, p<0.01), *Cidec* (p<0.001, p<0.001), *Plin4* (p<0.001, p<0.001), and *Fgf21* (p<0.01, p<0.01). Taken together, these data above suggest that XN and TXN antagonize PPARγ at the transcriptional level to block 3T3-L1 differentiation.

## XN and TXN antagonize ligand binding to PPARγ

Based on the inhibition of RGZ-induced adipocyte differentiation, and expression of PPARγ target genes, we postulated that XN and TXN bind to the PPARγ ligand-binding domain and interfere with agonist binding. To test this hypothesis, we first performed a competitive binding assay using a PPARγ time-resolved fluorescence resonance energy transfer (TR-FRET) assay. Both XN and TXN displaced a labeled pan-PPARγ ligand (Fluormone Pan-PPAR Green) in a dose-dependent manner with IC$_{50}$ values of 1.97 µM (*Figure 13B*) and 1.38 µM (*Figure 13C*), respectively. Oleic acid, the most

(Before XN section:) suppressed RGZ-induced differentiation in a dose-dependent manner. At 25 µM concentration, the RGZ-induced differentiation was largely blocked (*Figure 12B3,C3*), suggesting that XN and TXN may interfere or even compete with binding of RGZ to the PPARγ receptor.

**Table 5.** Thirteen genes[a] used to distinguish TXN transcriptome from HFD transcriptome.

| Ensemble ID | Gene name | Gene symbol | TXN vs. HFD (log$_2$ fold change) | p-value | FDR |
|---|---|---|---|---|---|
| 00000094793 | Major urinary protein 12 | *Mup12* | 2.65 | *0.000* | 0.011 |
| 00000033685 | Uncoupling protein 2 | *Ucp2* | −1.07 | *0.005* | 0.109 |
| 00000036390 | Growth arrest and DNA-damage-inducible 45 alpha | *Gadd45α* | −0.73 | 0.083 | 0.402 |
| 00000021226 | Acyl-CoA thioesterase 2 | *Acot2* | −1.33 | *0.000* | 0.003 |
| 00000030278 | Cell death-inducing DFFA-like effector c | *Cidec* | −2.41 | *0.000* | 0.006 |
| 00000043013 | One cut domain, family member 1 | *Onecut1* | 1.53 | *0.004* | 0.098 |
| 00000067219 | NIPA-like domain containing 1 | *Nipal1* | −0.63 | 0.197 | 0.567 |
| 00000035186 | Ubiquitin D | *Ubd* | −2.51 | *0.002* | 0.068 |
| 00000031842 | Phosphodiesterase 4C, cAMP specific | *Pde4c* | −0.00 | 0.996 | 0.999 |
| 00000026390 | Macrophage receptor with collagenous structure | *Marco* | 0.69 | 0.149 | 0.510 |
| 00000012187 | Monoacylglycerol O-acyltransferase 1 | *Mogat1* | −1.62 | *0.000* | 0.011 |
| 00000019942 | Cyclin-dependent kinase 1 | *Cdk1* | −1.53 | *0.009* | 0.139 |
| 00000046873 | Membrane-bound transcription factor peptidase | *Mbtps2* | −0.36 | 0.251 | 0.616 |

[a]Genes were ranked according to their *RReliefF* importance score using a multivariate filter technique (i.e., *RReliefF*) (***Chiesa et al., 2018***). Also shown is the log$_2$ fold changes, p-values, and FDR values when HFD-TXN samples were compared with HFD samples using edgeR package ***Robinson et al., 2010*** in R. Negative values indicate genes downregulated in the liver with TXN supplementation. Source files of data used for the analysis are available in ***Table 5—source data 1***.

The online version of this article includes the following source data for  Table 5:

**Source data 1.** Source files.This zip archive contains the following: (1) An Excel workbook named 'DEG_HFD_vs_TXN.xlsx' contains all differentially expressed genes identified. Genes listed in the table were highlighted in yell***Table 1***ow in the Excel workbook.

abundant FA ligand in the HFD diet (***Table 8***), had an IC$_{50}$ value of 16.6 μM. XN and TXN had similar IC$_{50}$ values as the PPARγ ligand PGZ, a drug used to improve glucose homeostasis and type 2 diabetes, and a natural ligand, arachidonic acid (***Chen et al., 2012***).

To obtain further insights into the interaction of XN and TXN with PPARγ, we analyzed the nature of binding between the PPARγ ligand-binding domain and XN/TXN using molecular docking to confirm the putative binding pose and position of XN/TXN and to estimate the relative binding affinities of various ligands for PPARγ. To verify the robustness of our docking protocol, resveratrol was re-docked into the bound structure of PPARγ, reproducing the binding pose and orientation found in the crystal structure of the complex (PDB ID: 4JAZ). The best docked position of TXN occupies the binding site of PPARγ, exhibiting many non-bonded interactions involving side chain atoms in Leu255, Phe264, Gly284, Cys 285, Arg288, Val339, Ile 341, Met348, and Met364 (***Figure 13D***). The side chains of His266, Arg280, and Ser342 and the main chain carbonyl oxygen atom of Ile281 are well positioned to make electrostatic/hydrogen bonds with the hydroxyl protons and oxygen atoms of the bound TXN molecule. We observed many of the same hydrophobic interactions in the simulated PPARγ-XN (***Figure 13E***) and PPARγ-oleic acid complexes, and potential electrostatic interactions between His266 and Glu343, or with Arg280 and XN or oleic acid, respectively. The relative binding affinities, ranked in decreasing value of their negative binding energies were, in order, TXN, XN, and oleic acid, consistent with the TR-FRET binding results.

## Discussion

### XN and TXN are effective in suppressing development of diet-induced steatosis

Low-cost natural products like XN are of particular interest for treating obesity and NAFLD due to their availability, safety, and efficacy. XN and its derivatives appear to function through multiple mechanisms of action, and this polypharmacological effect may enhance their effectiveness. Three studies propose that XN improves diet-induced hepatic steatosis by suppressing SREBP1c mRNA expression and SREBP activation (***Yui et al., 2014***; ***Miyata et al., 2015***; ***Takahashi and Osada,***

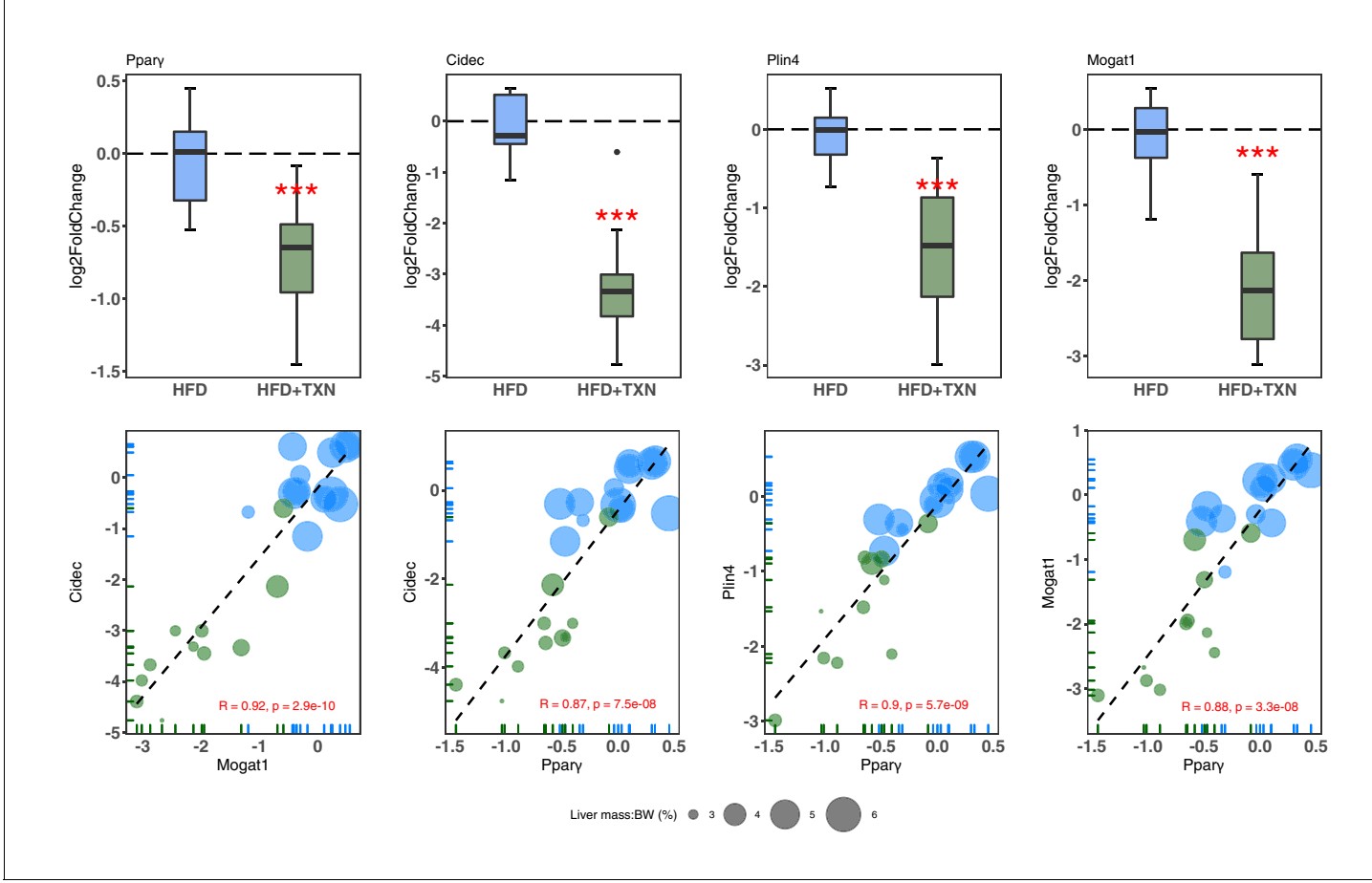

**Figure 10.** TXN-treated mice show significantly lower expression of PPARγ and target genes. Top panel: Reduction of HFD-induced *Pparg2*, *Cidec*, *Plin4*, and *Mogat1* expressions in the liver by TXN administration. Mice were sacrificed after 16 week of HFD (blue, n = 12) or HFD+TXN (dark green, n = 11) feeding. Liver tissues were harvested, and total RNA was extracted. Relative mRNA levels of selected genes were determined by real-time PCR. Gene expression is expressed in $\log_2$ fold change as quartiles. ***$p \leq 0.001$, t-test. Bottom panel: Pearson correlation between *Pparγ2* and *Cidec*, *Plin4* or *Mogat1* expression. Data are presented in $\log_2$ fold change; bubble size represents liver mass to BW ratio. • indicates sample outside value, which is >1.5 times the interquartile range beyond upper end of the box. Source files of data used for the analysis are available in *Figure 10—source data 1*. The online version of this article includes the following source data for figure 10:

**Source data 1.** Source files.

2017). We also observed a decrease in hepatic SREPB1c expression with TXN treatment. Others propose mechanisms include inhibiting pro-inflammatory gene expression (*Dorn et al., 2010*; *Mahli et al., 2019*), inducing AMPK activation in the liver and skeletal muscle (*Costa et al., 2017*), and enhancing FA oxidation (*Kirkwood et al., 2013*). In this study, using a combination of molecular, biochemical, biophysical, and bioinformatics approaches, we provide evidence for an additional novel mechanism by which XN and its derivative, TXN, can inhibit diet-induced hepatic steatosis through downregulation of hepatic FA uptake and lipid storage by binding to PPARγ in the liver and effectively antagonizing its actions.

We previously demonstrated that XN and TXN ameliorated DIO in C57Bl6/J mice with no evidence of liver injury (*Miranda et al., 2018*). Using the same animal model, we confirmed the phenotypic outcomes observed in the previous study (*Figure 2*, *Figure 2—figure supplement 1*). In this study and prior studies (*Miranda et al., 2018*), we noted a decrease in weight with treatment in the presence of similar caloric intake. Our metabolic cage data demonstrated energy expenditure increased with body mass, but a treatment effect was not identified. We hypothesize that changes in microbiota composition and bile acid metabolism, which can affect nutrient and energy harvesting, may explain the reduction in weight (*Wahlström et al., 2016*; *Zhang et al., 2020*) observed by

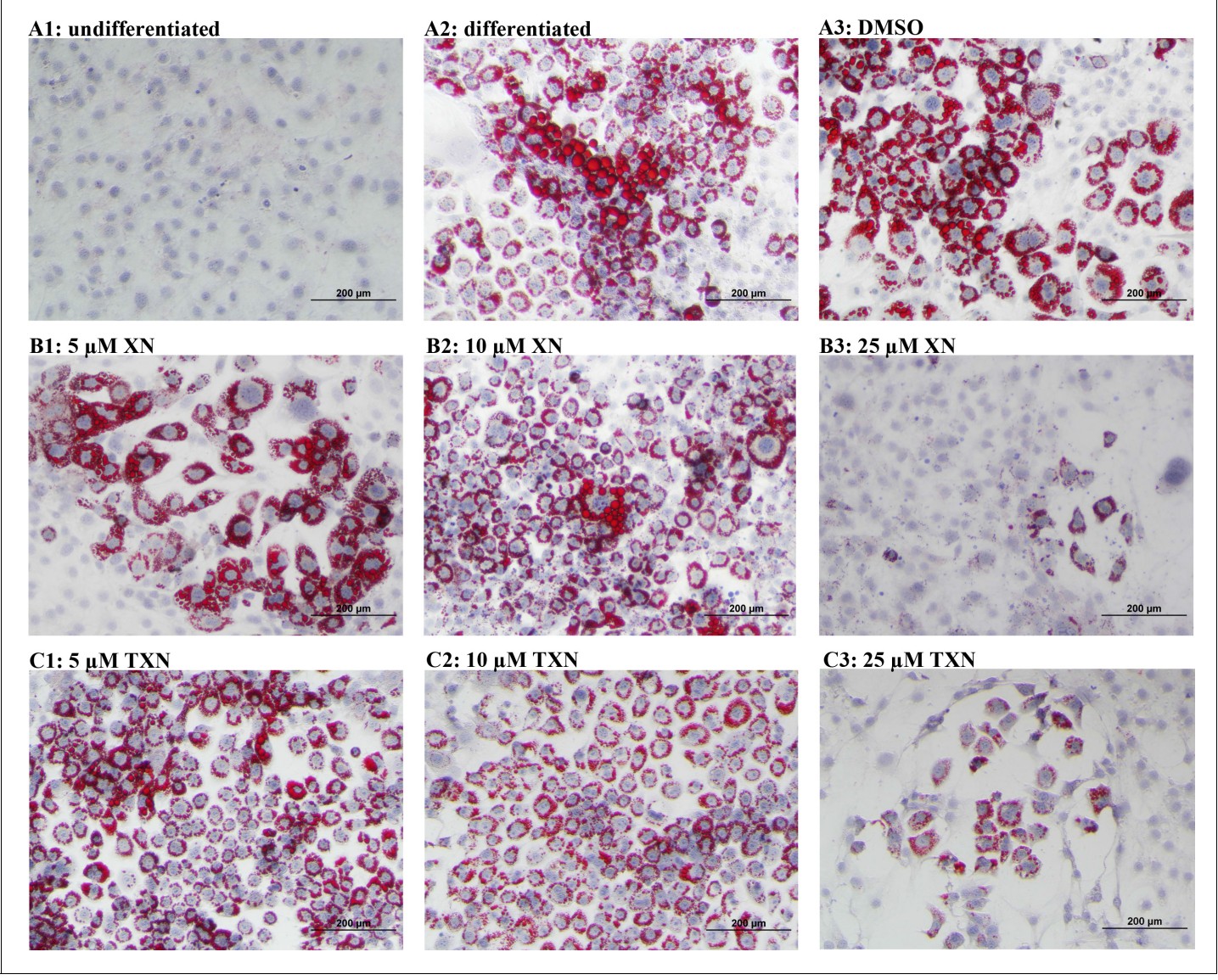

**Figure 11.** XN and TXN inhibit intracellular lipid accumulation in 3T3-L1 cells. 3T3-L1 cells (1 × 10⁶ per well) in 12-well plates were cultured with either DMEM (**A1**), differentiation medium (DM) (**A2**), DM plus DMSO (**A3**), DM plus 5 µM XN (**B1**), DM plus 10 µM XN (**B2**), DM plus 25 µM XN (**B3**), DM plus 5 µM TXN (**C1**), DM plus 10 µM TXN (**C2**), or DM plus 25 µM TXN (**C3**). Cells were stained with oil red O to identify lipids at day seven post-differentiation. DM: differentiation medium. Figshare link that contains raw images: https://doi.org/10.6084/m9.figshare.14744250.

treatment, but requires testing in future work. Furthermore, we demonstrated the effect of XN and TXN on the development and progression of diet-induced hepatic steatosis. Administration of 0.07% XN and 0.035% TXN significantly slowed the development and progression of hepatic steatosis during a 16 week high-fat feeding. We observed less macro- and microvesicular steatosis, significantly lower liver mass to BW ratio, decreased TAG accumulation, and significantly lower steatosis scores in the XN- and TXN-supplemented mice compared to their untreated HFD mice (*Figures 3B* and *6*). Four pathways generally maintain hepatic lipid homeostasis: uptake of circulating lipids, de novo lipogenesis (DNL), FA oxidation (FAO), and lipid export in very low-density lipoproteins (VLDL). These pathways are under tight regulation by hormones, nuclear receptors, and other transcription factors (*Bechmann et al., 2012*). Long-term dysregulation of one and/or multiple processes can lead to the development of NAFLD, obesity, type 2 diabetes, and other metabolic disorders.

To elucidate the mechanism of XN and TXN, we determined liver transcriptomic changes after 16 weeks of HFD feeding using RNA-seq. We observed significant changes in hepatic gene expression

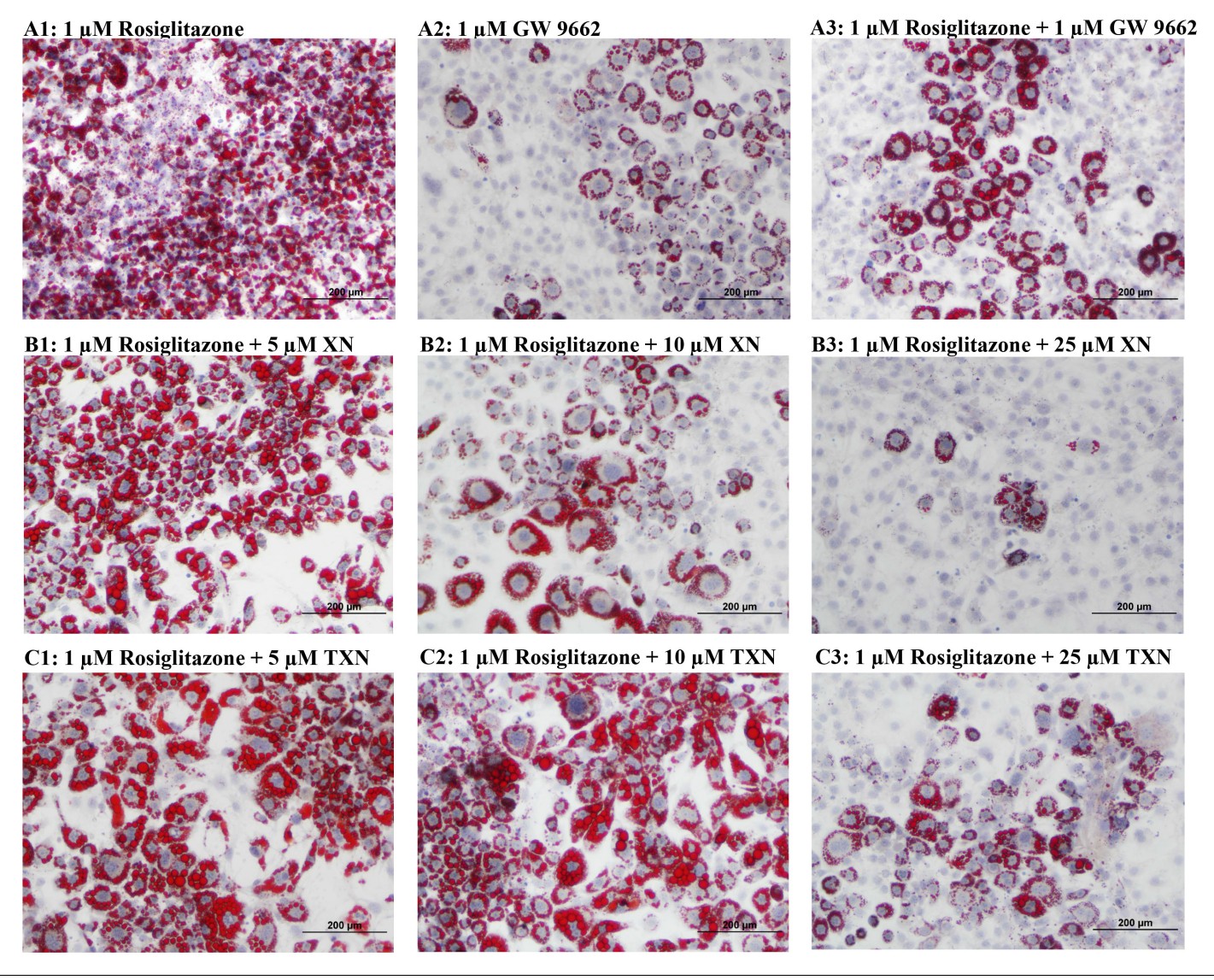

**Figure 12.** XN and TXN diminished the lipid accumulation in 3T3-L1 cells. 3T3-L1 cells ($1 \times 10^6$ per well) in 12-well plates were cultured with either DM plus 1 µM rosiglitazone (**A1**), DM plus 1 µM GW 9662 (**A2**), DM plus 1 µM rosiglitazone and 1 µM GW9662 (**A3**), DM plus 1 µM rosiglitazone and 5 µM XN (**B1**), DM plus 1 µM rosiglitazone and 10 µM XN (**B2**), DM plus 1 µM rosiglitazone and 25 µM XN (**B3**), DM plus 1 µM rosiglitazone and 5 µM TXN (**C1**), DM plus 1 µM rosiglitazone and 10 µM TXN (**C2**), or DM plus 1 µM rosiglitazone and 25 µM TXN (**C3**). Cells were stained with oil red O to identify lipids at day 7 post-differentiation. Figshare link that contains raw images: https://doi.org/10.6084/m9.figshare.14744250.

with TXN administration (*Figure 7B*). GO enrichment analysis of DEGs revealed that several biological processes were significantly downregulated by TXN treatment, including xenobiotic catabolism, FA metabolism, glucose metabolism, and regulation of lipid metabolism (*Figure 8*). Furthermore, KEGG pathway analysis of DEGs revealed that multiple biological pathways were downregulated in the livers of TXN-treated mice, including biosynthesis of unsaturated FAs, glutathione metabolism, amino sugar and nucleotide sugar metabolism, glycolysis and gluconeogenesis, FA elongation, and PPAR signaling pathways, suggesting that TXN rewired global hepatic lipid metabolism (*Figure 8*). There was a paucity of DEGs in the livers of mice supplemented with a high dose of XN even at an FDR cutoff of 0.4. This discrepancy might be due to reduced levels of XN in peripheral tissues as compared with TXN as we previously observed a 12-fold lower level of XN as compared with TXN in the liver (*Miranda et al., 2018*).

**Table 6.** Adipocyte gene expression at day seven post-differentiation.

| Gene | Log$_2$ (fold change) | | | | p-values vs. RGZ | | |
|------|------|------|------|------|------|------|------|
| | RGZ (cont) | RGZ + GW9662 | RGZ + XN | RGZ + TXN | RGZ + GW9662 | RGZ + XN | RGZ + TXN |
| Pparg2 | Ref. | −0.11 | −1.93 | −1.53 | 0.30 | <0.001 | <0.001 |
| Cd36 | | −0.18 | −9.10 | −4.36 | 0.25 | <0.001 | <0.001 |
| Fabp4 | | −0.12 | −7.94 | −4.08 | 0.43 | <0.001 | <0.001 |
| Mogat1 | | −0.11 | −4.16 | −3.59 | 0.42 | <0.001 | <0.01 |
| Cidec | | −0.18 | −10.10 | −4.46 | 0.40 | <0.001 | <0.001 |
| Plin4 | | −0.10 | −3.01 | −2.32 | 0.48 | <0.001 | <0.001 |
| Fgf21 | | 0.03 | −0.99 | −1.08 | 0.40 | <0.01 | <0.01 |

3T3-L1 differentiation was induced by IBMX, dexamethasone, insulin, and 1 µM RGZ plus the addition of 1 µM GW9662, 25 µM XN, or 25 µM TXN for 48 hr. After 48 hr, the old media was removed and fresh DMEM was replenished for continuing differentiation. Gene expression was measured at day 7 post-differentiation using qRT-PCR. ΔCT = CT(target gene) – CT(reference gene). ΔΔCT = ΔCT(treated sample) – ΔCT(untreated sample/control average). Fold change = $2^{-\Delta\Delta CT}$. Statistics were performed on ΔΔCT values. Source files of data used for the analysis are available in the **Table 6—source data 1**.

The online version of this article includes the following source data for Table 6:

Source data 1. Source files.This zip archive contains the following: (2) An Excel workbook named '7 days.xlsx' contains raw PCR cycle numbers, fold change, log(2) fold change, p-values, and how these are calculated.

To discover signature genes in the liver of mice treated with TXN, we applied a SVM classifier algorithm and extracted the most important features (genes) (**Figure 9**). Due to the limited number of samples in this study, we did not separate the data into training and testing sets for the construction of SVM. The caveat of this is that the learning model might not generalize well. Consistent with GO analysis, three of the eight significantly regulated genes – uncoupling protein 2 (*Ucp2*), cell death-inducing DFFA-like effector c (*Cidec*), and monoacylglycerol O-acyltransferase 1 (*Mogat1*) – are involved in lipid metabolism (**Table 5**). Notably, these genes are targets of PPARγ (**Bugge et al., 2010**; **Karbowska and Kochan, 2012**; **Wolf Greenstein et al., 2017**). qRT-PCR confirmed this finding (**Figure 10**) and suggested that TXN modulates PPARγ actions.

## XN and TXN are novel natural and synthetic PPARγ antagonists

PPARγ belongs to a super-family of nuclear receptors and just like other members, its activity requires ligand binding. PPARγ is highly expressed in white and brown adipose tissue, and to a lesser extent in the liver, kidney, and heart (**Zhu et al., 1993**; **Lee and Ge, 2014**). Because of its essential role in regulating adipogenesis and higher expression in the WAT, PPARγ has been a pharmacological target for drug development (**Lehmann et al., 1995**; **Lefterova et al., 2014**) in combating metabolic diseases such as insulin resistance and type 2 diabetes. Thiazolidinediones (TZDs), which include RGZ and PGZ, are the most widely investigated PPARγ agonists due to their strong insulin-sensitizing ability (**Henney, 2000**; **Soccio et al., 2014**). Studies show that the main action of TZDs occurs in adipocytes (**Chao et al., 2000**). In the liver, PPARγ plays a role in hepatic lipogenesis (**Sharma and Staels, 2007**). Multiple clinical trials using TZDs have observed significant improvement in hepatic steatosis and inflammation (**Ratziu et al., 2008**; **Ratziu et al., 2010**; **Sanyal et al., 2010**), suggesting additional actions of TZDs in non-adipocytes. Interestingly, PGZ is more effective in treating fatty liver disease than RGZ, the more potent PPARγ agonist (**Promrat et al., 2004**; **Ratziu et al., 2008**; **Ratziu et al., 2010**), suggesting that moderate binding is more effective. Unfortunate side effects of TZDs are weight gain (**Fonseca, 2003**), bone loss (**Schwartz and Sellmeyer, 2007**; **Schwartz, 2008**), edema, and increased risk of cardiovascular complications (**Nesto et al., 2004**; **Yang and Soodvilai, 2008**; **Bełtowski et al., 2013**), due to over-activation of PPARγ. Thus, there is great interest in identifying 'ideal' PPARγ modulators that are tissue specific with limited side effects.

An alternative strategy that aims to repress PPARγ has emerged in recent years (**Ammazzalorso and Amoroso, 2019**). The potential of reducing BW and improving insulin sensitivity suggests a possible clinical role of PPARγ antagonists in treating obesity and type 2 diabetes (**Yamauchi et al., 2001**; **Rieusset et al., 2002**; **Nakano et al., 2006**). Compared to agonists, researchers have identified only a few natural compounds that inhibit PPARγ, all of which have a

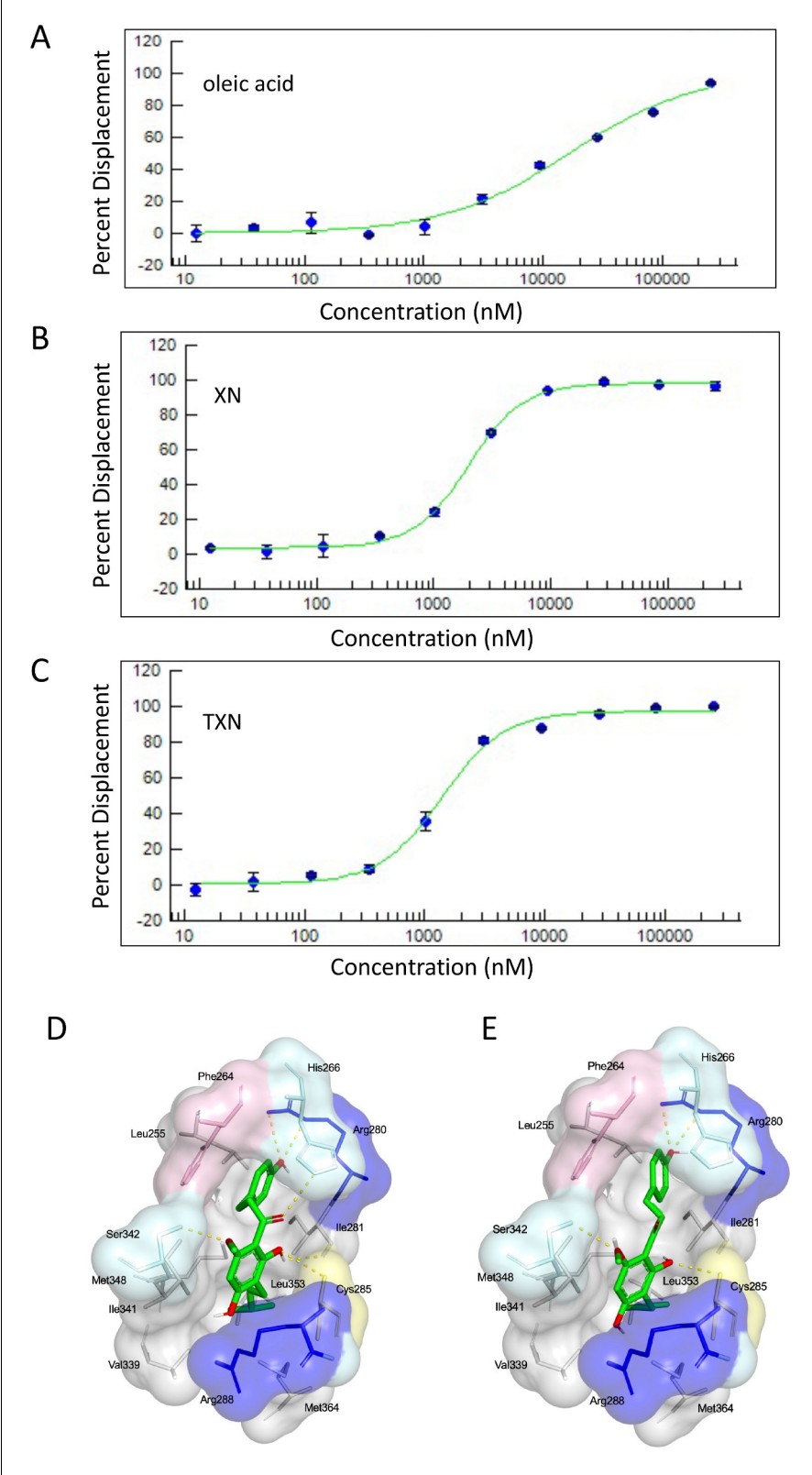

**Figure 13.** XN and TXN are ligands for PPARγ. A PPARγ nuclear receptor competitive binding assay based on time-resolved fluorescence resonance energy transfer (TR-FRET) was performed. The $IC_{50}$ values for each compound was determined by % displacement of a pan-PPARγ ligand. (**A**) Oleic acid $IC_{50}$ 16.6 μM. (**B**) XN $IC_{50}$ 1.97 μM. (**C**) TXN $IC_{50}$ 1.38 μM. Molecular docking studies show TXN and XN fit into the human PPARγ binding site. PPARγ residues containing atoms involved in hydrophobic interactions are shown. Yellow dashes indicate hydrogen bonds, amino acids colored as

*Figure 13 continued on next page*

*Figure 13 continued*
hydrophobic (gray), aromatic (pink), polar (cyan), basic (blue), or cysteine (yellow). (D) TXN and (E) XN. Source files of data used for the analysis are available in *Figure 13—source data 1*.

The online version of this article includes the following source data for figure 13:

**Source data 1.** Source files: an Excel file named 'SSBN12209_57828_10-point Titration_Inhibition_Results.xls' containing results from ThermoFisher PPARγ nuclear receptor competitive binding assay.

moderate binding affinity for PPARγ receptor and can inhibit adipogenesis, obesity, and/or hepatic steatosis. These include resveratrol (*Calleri et al., 2014*), 7-chloroarctinone-b isolated from the roots of *Rhaponticum uniflorum* (*Li et al., 2009*), tanshinone IIA from the roots of *Salvia miltiorrhiza* (danshen) (*Gong et al., 2009*), astaxanthin from red-colored aquatic organisms (*Jia et al., 2012*), protopanaxatriol extracted from *Panax ginseng* roots (*Zhang et al., 2014*), foenumoside B from the herbal plant *Lysimachia foenum-graecum* (*Kwak et al., 2016*), and betulinic acid, a pentacyclic triterpene found in the bark of several plants (*Brusotti et al., 2017*; *Ammazzalorso and Amoroso, 2019*).

Several lines of evidence presented in this study support the hypothesis that XN and TXN are also PPARγ antagonists. First, using the 3T3-L1 cell model for PPARγ-mediated adipogenesis, we demonstrated that XN and TXN significantly and strongly suppressed RGZ-induced adipocyte differentiation and adipogenesis by day 7 (*Figure 12*). Consistent with a decrease in lipid accumulation, PPARγ target genes were also significantly downregulated in XN- and TXN-treated cells (*Table 6*). The PPARγ antagonist, GW9662, did not significantly affect target gene expression of *Pparg*, even though it inhibited differentiation (*Figure 12A2-3*). In our experiments, we used a significantly lower concentration of GW9662 than used by others that ranged from 3 to 25 times higher, and this difference could explain our results (*Park et al., 2008*; *Kim et al., 2011*; *Sankella et al., 2016*). Second, the PPARγ nuclear receptor competitive binding assay showed that XN and TXN have a moderate binding affinity of 1.97 µM and 1.38 µM, respectively (*Figure 13*). Lastly, consistent with the competitive binding assay, simulated molecular docking indicated that XN and TXN can interact with the ligand-binding domain of PPARγ like other known ligands and potentially form hydrogen bonds with His266, Arg280, Ser342, and Ile281, in addition to many non-bonded interactions (*Figure 13D,E*). Moreover, the predicted binding model reveals that the interactions between XN, TXN, and the PPARγ ligand-binding domain resembles those observed between PPARγ and resveratrol, a dietary polyphenol that is also a PPARγ antagonist (*Calleri et al., 2014*). Our findings are consistent with XN and TXN functioning as PPARγ antagonists and now offer a mechanistic explanation for prior

**Table 8.** Fatty acid composition (% of the total fat) of the low-fat diet (LFD) and high-fat diet (HFD).

| | % of the total fat | | g/kg diet | |
|---|---|---|---|---|
| **Fatty acids** | **LFD** | **HFD** | **LFD** | **HFD** |
| 14:0 Myristic | 0.7 | 1.4 | 0.29 | 4.75 |
| 16:0 Palmitic | 17.0 | 24.2 | 7.28 | 84.34 |
| 16:1 Palmitoleic | 1.5 | 3.1 | 0.65 | 10.76 |
| 18:0 Stearic | 8.3 | 12.3 | 3.56 | 42.92 |
| 18:1 Oleic | 32.2 | 42.1 | 13.76 | 146.95 |
| 18:2 Linoleic | 35.2 | 14.9 | 15.04 | 51.89 |
| 18:3 Linolenic | 5.0 | 2.1 | 2.14 | 7.27 |
| SFAs | 26.0 | 37.9 | 11.13 | 132.01 |
| MUFAs | 33.7 | 45.2 | 14.41 | 157.71 |
| PUFAs | 40.2 | 17.0 | 17.18 | 59.16 |
| Total n-6 PUFA | 35.2 | 14.9 | 15.04 | 51.89 |
| Total n-3 PUFA | 5.0 | 2.1 | 2.14 | 7.27 |

Abbreviations: SFA: saturated fatty acids; MUFAs: monounsaturated fatty acids; PUFAs: polyunsaturated fatty acids; n-6: omega-6 fatty acids; n-3: omega-3 fatty acids.

observations that XN impaired adipocyte differentiation (*Yang et al., 2007*; *Mendes et al., 2008*; *Samuels et al., 2018*).

One of the many side effects observed from TZD therapy is weight gain. TZDs primarily mediate their effects in adipose tissue by PPARγ activation that stimulates adipocyte differentiation and increases the efficiency of uptake of circulating non-esterified FAs by adipocytes (*Rosen and Spiegelman, 2006*). Interestingly, in this study, we observed a significant decrease in overall, sWAT, and mWAT fat mass in HXN- and TXN-treated mice (*Figure 3A*, *5AC*), yet a slight increase in the eWAT fat mass (*Figure 5B*). Prior studies have reported that the expandability of eWAT in male mice is an indicator of metabolic health. Mouse sWAT and mWAT will continue to expand with BW, whereas eWAT expansion diminishes after mouse BW reaches about 40 g (*van Beek et al., 2015*). Our data suggest that HXN- and TXN-treated mice have capacity to expand eWAT, whereas HFD-fed untreated mice do not, which seems to direct the development of metabolic disorders. In our previous study, we demonstrated that XN and TXN accumulates primarily in the liver with significantly lower levels in the muscle (*Miranda et al., 2018*). We could not detect XN or TXN in the WAT of these mice (data not shown). The levels of XN and TXN in the liver (TXN > HXN > LXN) and the absence of both compounds in the WAT suggest that these compounds antagonize PPARγ in the liver and not in the WAT, therefore, minimizing the side effect of weight gain observed with TZDs that are PPARγ agonists.

During a long-term HFD feeding, PPARγ and its target genes are upregulated to compensate for the lipid overflow in the liver. Namely, genes associated with lipid uptake and trafficking (*Lpl*, *Cd36*, *Fabp4*), TAG synthesis (*Fasn*, *Scd1*, *Mogat1*), and formation of lipid droplets for storage (*Cidec/Fsp27*, *Plin4*) (Supplement_File_B). The result is excessive lipid accumulation in the liver, leading to hepatic steatosis. This was observed with PPARγ overexpression in hepatocytes in *ob/ob* mice (*Rahimian et al., 2001*). We propose that TXN added to a HFD antagonizes PPARγ action in the liver potentially by physically interacting with PPARγ receptors as indicated in the molecular docking studies (*Figure 13DE*) and, therefore, reduces PPARγ transcriptional activity and expression of the aforementioned target genes. Several in vivo studies support our findings. Hepatocyte- and macrophage-specific PPARγ deficiency protects Lep < *ob/ob*> mice from hepatic steatosis (*Matsusue et al., 2003*; *Morán-Salvador et al., 2011*); knockdown of *Mogat1* in the liver significantly attenuates hepatic steatosis after 12 weeks HFD feeding (*Lee et al., 2012*); and restoration of *Cidec/Fsp27* in Lep < *ob/ob*> liver-specific *Pparg* knockout mice promotes hepatic steatosis (*Matsusue et al., 2008*). The role for *Plin4* in hepatic steatosis is limited, but it may affect TAG accumulation during HFD feeding (*Griffin, 2017*). Ablation of *Pparg* in murine myeloid cells increased insulin resistance (*Souza et al., 2020*) and ablation in macrophages and hepatic stellate cells, but not hepatocytes increased inflammation (*Morán-Salvador et al., 2013*). Nevertheless, we did not observe either of these conditions in our study. In contrast, TXN did not promote hepatic inflammation (*Table 2*) but improved glucose clearance (*Figure 2—figure supplement 1*). We postulate complete the absence of PPARγ is quite different from modulating its activity through agonists and antagonists, and this may explain the differences noted in some of these cell-specific knockout studies and our findings.

As we discussed earlier, antagonizing PPARγ action is likely an additional mechanism by which XN and TXN suppress diet-induced NAFLD. Other possible mechanisms may play a role as well. Gut microbiota dysbiosis has been observed in obesity and type 2 diabetes, which are diseases strongly associated with NAFLD. Mouse studies and fecal transplantation experiments have demonstrated a causal role of gut microbiota in the development of NAFLD (*Henao-Mejia et al., 2012*). We previously reported that both XN and TXN drastically changed gut microbiota composition in C57Bl6/J male mice, accompanied with a significant change in the fecal bile acid composition (*Zhang et al., 2020*). Specifically, administration of XN and TXN decreased intestinal microbiota diversity and abundance, altered bile acid metabolism, and reduced inflammation. Changes in the gut microbiota and bile acid metabolism may also explain, in part, the improvements in MetS and NAFLD, but requires investigation in future studies.

Legette et al. reported feeding XN to Zucker fa/fa rats for 6 weeks significantly lowered BW gain and plasma glucose levels only in male, but not female rats (*Legette et al., 2013*). This gender difference in response to XN is not unique as similar findings were observed for other flavonoids (*Camper-Kirby et al., 2001*; *Blair et al., 2002*; *Guo et al., 2005*). Consistent with a prior study (*Zhou et al., 2009*), we found the expression of numerous major urinary protein (*Mup*) genes

reduced by a HFD as compared to the LFD; however, the expression of these same genes were induced by TXN administration (*Table 3*). MUPs are unique members of the lipocalin super-family produced by the liver and secreted into urine primarily in males (*Zhou and Rui, 2010*). They function in the urine as pheromones in chemical communication and as metabolic signals regulating glucose and lipid metabolism in individual animal (*Zhou and Rui, 2010*). Because adult male mice secrete significantly more MUPs than females, this finding may explain why female mice do not respond to XN and TXN like their male counterparts. TXN treatment also beneficially modulated expression of lipocalin members, Lcn2 and ApoMm (*Wang et al., 2007*; *Milner et al., 2009*; *Auguet et al., 2013*; *Yang et al., 2019*). To elucidate the role these changes in gene expression play in gender-specific responses to TXN requires additional research.

To maintain energy homeostasis, proper crosstalk between metabolically active tissues is essential (*Stern et al., 2016*). In NAFLD (and MetS in general), these tissues often present a chronic low-grade inflammation characterized by the recruitment of pro-inflammatory cells, cytokines, and acute-phase proteins (*Lackey and Olefsky, 2016*; *Wang et al., 2021*). Previously we reported that supplementation with TXN decreased chronic inflammation with reduced expression of major pro-inflammatory cytokines *Il6* and *Tnfα* in WAT and to a smaller extent in the liver (*Zhang et al., 2020*). We also observed a decrease in *Ccl2*, a chemotactic factor involved in the recruitment of monocytes, and macrophage marker F4/80 in WAT, suggesting that TXN may protect WAT from macrophage infiltration (*Zhang et al., 2020*).

In conclusion, we demonstrated the dose of TXN given in the diet is very effective in suppressing the development and progression of diet-induced hepatic steatosis in mice. TXN appears more effective in vivo than XN perhaps due to significantly higher levels of TXN in the liver, but XN can slow progression of the condition at a higher dose. At the dose used for TXN, we have not observed adverse events in our prior or current studies (*Miranda et al., 2018*). In future work, we would consider testing lower and higher doses for safety and efficacy and pursuing pharmacokinetic studies such as those already done with XN (*Legette et al., 2012*; *Legette et al., 2014*). While XN and TXN are effective preventative approaches in rodents, in future studies we are interested in determining if these compounds can treat existing obesity. We provide evidence that XN and TXN act as novel, natural, and synthetic antagonists of PPARγ that bind with a similar affinity as the agonist PGZ. Our findings support further development of XN and TXN as novel, low-cost therapeutic compounds for diet-linked hepatic steatosis with fewer negative side effects than current drugs (e.g., reduced adipose tissue expansion). Additionally, the structures of XN and TXN could serve as scaffolds for the synthesis of more effective compounds to treat NAFLD and other fatty liver diseases. These findings also raise the possibility of testing XN and TXN in combination with other PPARγ ligands in treating obesity and metabolic syndrome. Although these results are encouraging, further studies are required to clarify possible use in humans for the prevention and treatment of diet-linked hepatic steatosis.

## Materials and methods

### Animals and diets

Studies were performed using 8-week-old SPF male C57Bl/6J mice obtained from The Jackson Laboratory (Bar Harbor, ME). Upon arrival, 60 mice were housed individually in ventilated cages in a controlled environment (23 ± 1℃, 50–60% relative humidity, 12 hr daylight cycle, lights off at 18:25 hr) with food and water ad libitum. After acclimating mice for 1 week on a normal-chow diet (PicoLab Rodent Diet 20, 5053, TX) followed by 2 weeks on a low-fat control diet (LFD; Dyets Inc, Bethlehem, PA), they were randomly assigned (restricted) to five groups (n = 12/group). The sample size of 12 mice per treatment group was based on previous published studies (*Miranda et al., 2016*; *Miranda et al., 2018*). The groups were fed either a LFD, HFD, HFD + 0.035% XN (LXN), HFD + 0.07% XN (HXN), or HFD + 0.035% TXN (TXN). XN and TXN (both of purity >99%) were provided by Hopsteiner, Inc (New York, NY). The chemical structures of XN and TXN, a detailed diet composition, and FA composition are available in *Figure 1*, *Table 7*, and *Table 8*, respectively.

BW gain and food intake of individual mice were assessed once per week. Body composition was determined at the end of the feeding using a Lunar PIXImus 2 Dual Energy X-ray Absorptiometer (DXA) scan (Madison, WI). After 16 weeks of feeding the control and test diets, mice were fasted for

**Table 7.** Composition of diets[a].

| | HFD | HFD + LXN | HFD + HXN | HFD + TXN | LFD |
|---|---|---|---|---|---|
| *Ingredient (g/100 g)* | | | | | |
| Casein | 2.58 | 2.58 | 2.58 | 2.58 | 1.89 |
| L-Cystine | 0.04 | 0.04 | 0.04 | 0.04 | 0.03 |
| Sucrose | 0.89 | 0.89 | 0.89 | 0.89 | 0.89 |
| Cornstarch | 0.00 | 0.00 | 0.00 | 0.00 | 4.02 |
| Cellulose | 0.54 | 0.54 | 0.54 | 0.54 | 0.47 |
| Dyetrose | 1.62 | 1.62 | 1.62 | 1.62 | 1.62 |
| Soybean oil | 0.32 | 0.32 | 0.32 | 0.32 | 0.24 |
| Lard | 3.17 | 3.17 | 3.17 | 3.17 | 0.19 |
| Mineral Mix #210088 | 0.13 | 0.13 | 0.13 | 0.13 | 0.10 |
| Dicalcium phosphate | 0.17 | 0.17 | 0.17 | 0.17 | 0.12 |
| Calcium carbonate | 0.07 | 0.07 | 0.07 | 0.07 | 0.05 |
| Potassium citrate $H_2O$ | 0.21 | 0.21 | 0.21 | 0.21 | 0.16 |
| Vitamin mix #300050 | 0.13 | 0.13 | 0.13 | 0.13 | 0.10 |
| Choline bitartrate | 0.03 | 0.03 | 0.03 | 0.03 | 0.02 |
| Test compound | 0.00 | 0.003 | 0.006 | 0.003 | 0.00 |
| OPT | 0.10 | 0.10 | 0.10 | 0.10 | 0.10 |
| *Composition (kcal%)* | | | | | |
| Protein | 20 | 20 | 20 | 20 | 20 |
| Carbohydrates | 20 | 20 | 20 | 20 | 70 |
| Lipids | 60 | 60 | 60 | 60 | 10 |
| *Energy density (kcal/g)* | 5.12 | 5.12 | 5.12 | 5.12 | 3.55 |

[a]LXN provides 0.035% xanthohumol (XN), HXN (0.07% XN), and 0.035% TXN per day. The test compounds were dissolved in an isotropic mixture of oleic acid: propylene glycol: Tween 80 (OPT) 0.9:1:1 by weight before incorporation into the diets. All diets were purchased from Dyets Inc, Bethlehem, PA.

6 hr during the dark cycle, anaesthetized in chambers saturated with isoflurane, and then euthanized by cardiac puncture followed with cervical dislocation. Blood was collected in syringes containing 2 IU of heparin and centrifuged to separate plasma from cells. The liver and sWAT, mWAT, and eWAT fat pads were carefully collected and weighed. To avoid batch effect due to difference in hours of fasting, mice were randomized (restricted), and treatment information was masked before sacrifice. The Institutional Animal Care and Use Committee (IACUC) at Oregon State University approved all animal work (ACUP 5053). All animal experiments were performed in accordance with the relevant guidelines and regulations as outlined in the Guide for the Care and Use of Laboratory Animals.

## Glucose tolerance

Glucose tolerance tests were conducted after 9 weeks of feeding the experimental diets. Mice were fasted for 5 hr (during light cycle) and weighted at the end of fasting prior to baseline (t = 0 min) blood glucose testing. Mice then received a glucose bolus (2 g/kg; 20% glucose solution, w/v) through i.p. injection. Circulating glucose levels were measured with AlphaTRAK2 blood glucose test strips and AlphaTRAK2 glucometer with cat setting (Zoetis Inc, MI) at 0 (before the injection), 15 min, 30 min, 1 hr, and 2 hr after the injection by tail puncture with a 28-gauge lancet.

## Liver histology

Liver (~100 mg) was freshly collected from mice and immediately fixed overnight in 10% neutrally buffered formalin, paraffin embedded, sectioned, and stained with hematoxylin-eosin (Veterinary Diagnostic Laboratory, Oregon State University, OR). Each slide contained two liver sections that were examined using a Leica microscope at 100× magnification. Representative images were taken

at 100× magnification from the subjectively least and most severely affected areas ensuring representation of all zones of the hepatic lobule. Steatosis was objectively quantified as percent surface area occupied by lipid vacuoles using ImageJ for image analysis (NIH; imagej.nih.gov/ij/index.html) as previously published (*Garcia-Jaramillo et al., 2019*).

## Energy expenditure

Indirect calorimetry measurements were based on an open respirometer system. From week 10, mice were housed individually in Promethion Line metabolic phenotyping chambers (Sable Systems International, Las Vegas, NV) and maintained on a standard 12 hr light/dark cycle for 3 days. The system consisted of 10 metabolic cages, each equipped with food and water hoppers connected to inverted laboratory balances for food intake monitoring; both food and water were available ad libitum. Spontaneous physical activity (SPA) was quantified via infrared beam breaks in X and Y axes and included locomotion, rearing, and grooming behaviors (BXY-R, Sable Systems International). All raw data from all sensors and analyzers were stored every second. Air within the cages was sampled through micro-perforated stainless-steel sampling tubes located around the bottom of the cages, above the bedding. Ambient air was passed through the cages (2 l/min), and gases were sampled continuously for each cage, allowing the simultaneous acquisition of metabolic data every second, for all cages in the system (*Lighton and Halsey, 2011*). The energy expenditure was estimated from oxygen consumption ($VO_2$) and carbon dioxide production ($VCO_2$) rates by the Promethion system using the Weir formula (*Weir, 1949*).

## Liver tissue RNA extraction and library preparation

Freshly dissected liver tissue was flash frozen in liquid $N_2$ and then stored at −80℃. Total RNA was isolated using the Direct-zol RNA Miniprep Plus kit as instructed (Zymo Research, Irvine, CA). RNA concentrations were quantified using the Qubit 1.0 Fluorometer and the Qubit RNA BR Assay kit (Thermo Fisher Scientific, Waltham, MA). RNA purity and integrity were evaluated using a Bioanalyzer RNA 6000 Nano chip (Agilent Technologies, Santa Clara, CA). Samples ranged from medium to high RNA quality (RIN 5.9–8.3), and samples with different RIN values showed similar RNA-seq qualities.

Each library was prepared with 325 ng total RNA using the Lexogen QuantSeq 3'mRNA-Seq Library Prep Kit-FWD for Illumina sequencing according to the manufacturer's instructions (Lexogen GmbH, Vienna, Austria). Briefly, library preparation was started by oligo(dT) priming, with primers already containing the Illumina-compatible linker sequence for Read 2. After first-strand synthesis, the RNA was removed before random primers that contained the corresponding Illumina-compatible linker sequence for Read 1 initiated the second-strand synthesis. Second-strand synthesis was followed by a magnetic bead-based purification step. The libraries were PCR amplified introducing sequences required for cluster generation and i7 and i5 dual indices (Lexogen i7 six nt Index Set and Lexogen i5 six nt Unique Dual Indexing Add-on Kit) for 16–20 PCR cycles with the optimal number predetermined by qPCR with the PCR Add-on Kit for Illumina (Lexogen GmbH). After a second magnetic bead-based purification, libraries were quantified using the Qubit dsDNA HS Assay Kit (Thermo Fisher Scientific) and sized using an Agilent High Sensitive D5000 Screen Tape (Agilent Technologies) to determine molarity. Equal molar amounts of the libraries were multiplexed and then sequenced on an Illumina Hiseq3000 platform (Illumina, San Diego, CA) at the Center for Genome Research and Biocomputing, Oregon State University using single-end sequencing with 100 bp reads. Approximately 6.6 million reads were obtained per liver sample.

## Sequence alignment and gene counts

Adaptors and low-quality tails were trimmed, and ribosomal rRNA contaminations were removed using BBDuk from the BBTools toolset (*Bushnell, 2014*). As recommended by the manufacturer (Lexogen GmbH), a Phred score of 10 and a read length of 20 were used as the minimum cutoff prior to data analysis (https://www.lexogen.com/quantseq-data-analysis/). Using a splice-aware aligner STAR (*Dobin et al., 2013*) (version 37.95), cleaned reads were then mapped against the GRCm38 primary assembly of the *Mus musculus* genome (version mm10, M22 release) (ftp://ftp.ebi.ac.uk/pub/databases/gencode/Gencode_mouse/release_M22/GRCm38.primary_assenbly.genome.fa.gz), with the annotation file of the same version (ftp://ftp.ebi.ac.uk/pub/databases/gencode/Gencode_mouse/

release_M22/gencode.vM22.annotation.gtf.gz), both from the GENCODE project (*Frankish et al., 2019*). On average, over 81% of the reads were uniquely mapped for each sample. Downstream analyses were based on uniquely aligned reads.

To generate count matrices from bam files, the summarizeOverlaps function from the GenomicAlignments package (v1.26.0) was used (*Lawrence et al., 2013*). The location of the exons for each gene was obtained from a transcript database (TxDb) using the makeTxDbFromGFF function from the GenomicFeatures package (version 1.42.1), with a pre-scanned GTF file used in the mapping step. Genes were then annotated with the R package *Mus musculus* (version 1.3.1) (*Team, 2016*).

## Identification of DEGs

R package edgeR (version 3.26.8) was used to detect differential change in gene expression among mice on different diets (*Robinson et al., 2010*). Genes expressed in at least nine samples were retained using the filterByExpr function in edgeR. Unannotated genes, pseudogenes, and ribosomal RNA genes were also removed from downstream analyses. Gene counts were then normalized with the default TMM (trimmed mean of M-values) method (*Robinson and Oshlack, 2010*) provided by edgeR. To account for both biological and technical variability, an overdispersed Poisson model and an Empirical Bayes method were used to moderate the degree of overdispersion across transcripts. Genes with an FDR threshold < 0.4 were used for heatmap and volcano plot analyses, whereas genes with an FDR threshold < 0.05 were used in GO and pathway enrichment analysis.

## GO and pathway enrichment analyses

GO and KEGG pathway enrichment analysis was conducted using Enrichr (http://amp.pharm.mssm.edu/Enrichr) (*Chen et al., 2013*; *Kuleshov et al., 2016*). Genes with an FDR threshold < 0.05 were analyzed with GO biological process 2018 and KEGG 2019 Mouse databases. Full tables can be found in the supplementary material (Supplement_File_A).

## Classification of RNA-seq data

Gene selection and normalization were performed using the R package DaMiRseq 1.2.0 (*Chiesa et al., 2018*). To distinguish TXN-fed samples from HFD control samples, we used a correlation cutoff of 0.4 for the partial least-squares feature selection (FSelect), and the default correlation coefficient for the redundant feature removal (FReduct).

## Cell culture

Murine 3T3-L1 preadipocytes were obtained from ATCC (Rockville, MD). We did not note mycoplasma contamination. Prior to treatments, cells were maintained in basic media, which consisted of high-glucose DMEM supplemented with 1% penicillin-streptomycin and 10% heat-inactivated FBS (Hyclone, Logan, UT). The cells were allowed to reach full confluence for 2 days. Differentiation was induced by the addition of 0.5 μM IBMX (Sigma-Aldrich, St. Louis, MO), 0.25 μM dexamethasone (Sigma-Aldrich), and 10 μg/ml insulin (Sigma-Aldrich) plus the addition of treatment compounds XN or TXN. After 48 hr, media was removed and fresh DMEM was replenished for continuing differentiation. To observe XN and TXN's effects on 3T3-L1 adipocyte differentiation, different concentrations were selected based on dose-response experiments to identify the dose that maximized effectiveness while minimizing cell toxicity.

## MTT cell viability assay

For cell viability experiments using the MTT assay, 3T3-L1 fibroblasts were seeded in 96-well plates at a density of 15,000 cells per well in 200 μl of DMEM medium supplemented with 10% FBS, 1% glutamine, 1 mM of sodium pyruvate, 100 units/ml penicillin, and 100 μg/ml streptomycin. After incubating 48 hr with various concentrations of XN or TXN at 37°C in 5% $CO_2$ atmosphere, the culture medium was removed and a solution of MTT [3-(4,5-dimethylthiazol-2-yl)−2,5-diphenyltetrazolium bromide], 0.5 mg/ml in complete culture medium, was added to each well. The cells were incubated with MTT for 3 hr at 37°C and then the MTT medium was removed before adding acidified isopropanol to each well. The cells were shaken for 10 min in an orbital shaker before reading the absorbance at 570 nm using a Microplate Reader (SpectraMax 190, Molecular Devices, Sunnyvale,

CA). Cell viability of compound-treated cells was calculated as percent absorbance of vehicle-treated control cells.

## Oil red O staining

Cells were washed twice with phosphate-buffer saline (PBS) and then fixed with 10% formalin for 30 min. Cells were then washed with ddH$_2$O followed by 60% isopropanol. A 0.4% stock solution of Oil Red O (Sigma-Aldrich) in isopropanol was diluted 3:2 (Oil red O:ddH$_2$O) for a working solution. To determine intracellular lipid accumulation, fixed cells were incubated for 30–60 min at room temperature on a rocker with the Oil red O working solution. After incubation, cells were washed with ddH$_2$O and imaged using microscopy.

## Adipocyte gene expression by RT-qPCR

Total RNA was isolated as described above, dissolved in RNase-free water, and stored at −80°C. For RT-PCR experiments, cells were grown in six-well plates and treated with XN and TXN at 25 µM concentration and differentiation medium after confluence for 2 days. Gene expression was measured from cells at 7 d post treatment. RNA (0.25 µg) was converted to cDNA using iScript reverse transcriptase and random hexamer primers (Bio-Rad Laboratories), according to the manufacturer's recommendations. PCRs were set up as described previously (*Gombart et al., 2005*). All the threshold cycle number (CT) were normalized to Ywhaz reference gene. PrimeTime Std qPCR assays were purchased from IDT (*Table 9*). ΔCT = CT(target gene) – CT(reference gene). ΔΔCT = ΔCT(treated sample) – ΔCT(untreated sample/control average). Statistics were done on ΔΔCT values.

## Time-resolved fluorescence resonance energy transfer

To determine the binding affinity of XN and TXN to PPARγ, a Lanthascreen TR-FRET PPARγ competitive binding assay was performed by Thermo Fisher Scientific (cite manual) (Lanthascreen, Invitrogen). A terbium-labeled anti-GST antibody binds to a GST-PPARγ-ligand-binding domain fusion protein in which the LBD is occupied by a fluorescent pan-PPAR ligand (Fluormone Pan-PPAR Green). Energy transfer from the antibody to the ligand occurs and a high TR-FRET ratio (emission signal at 520 nm/495 nm) is detected. When a test compound displaces the ligand from PPARγ-LBD, a decrease in the FRET signal occurs and a lower TR-FRET ratio is detected (*Corporation, 2008*). For each compound (XN, TXN, or oleic acid), a 10-point serial dilution (250,000–12.5 nM) was tested. Binding curves were generated by plotting percent displacement versus log concentration (nM), and IC$_{50}$ values were determined using a sigmoidal dose response (variable slope).

**Table 9.** Primer probe information.

| Gene name | IDT assay name | RefSeq number |
| --- | --- | --- |
| Cd36 | Mm.PT.58.12375764 | NM_007643 |
| Cidec/Fsp27 | Mm.PT.58.6462335 | NM_178373 |
| Fabp4 | Mm.PT.58.43866459 | NM_024406 |
| Fgf21 | Mm.PT.58.29365871.g | NM_020013 |
| Il6 | Mm.PT.58.10005566 | NM_031168 |
| Lpl | Mm.PT.58.46006099 | NM_008509 |
| Mogat1 | Mm.PT.58.41635461 | NM_026713 |
| Pparg2 | Mm.PT.58.31161924 | NM_011146 |
| Plin4 | Mm.PT.58.43717773 | NM_020568 |

## Molecular docking simulations for XN and TXN into the PPARγ ligand-binding domain

To estimate the binding mode of XN and TXN to PPARγ, molecular docking simulations were performed using AutoDock Vina (*Trott and Olson, 2010*). Structural models of XN and TXN were built using OpenBabel to convert the isometric SMILES descriptor for XN to a PDB formatted file, which was subsequently modified using PyMOL (The PyMOL Molecular Graphics System, Version 1.7.4.5, Schrödinger, LLC) to obtain a PDB file for TXN. The solved structure of PPARγ bound to the antagonist resveratrol (PDB ID: 4JAZ) was used as the receptor model. The PDBQT files for the receptor and the resveratrol, XN, TXN, and oleic acid ligands were generated using MGLTools-1.5.7rc1 (*Morris et al., 2009*). The PPARγ receptor was kept rigid during all docking experiments, and the center and size ($20 \times 20 \times 20$ Å$^3$) of the docking box was positioned to cover the entire ligand-binding site of PPARγ. All rotatable torsion angles in the ligand models were allowed to be active during the docking simulations. Twenty docking poses were generated for each simulation, and the conformation with the lowest docking energy was chosen as being representative.

## Statistical analysis

Analysis of variance procedures for continuous data and Fisher's exact test for binary data were used for statistical comparisons. p-values of orthogonal a priori comparisons of the HFD control group versus each of the supplement groups are shown in the corresponding tables and figures. Additional details of statistical analyses are described in the corresponding figure legends.

## Acknowledgements

We thank Jamie Pennington, Scott Leonard, and Dr. Wenbin Wu for their assistance, Dr. Edward Davis for bioinformatics support and Anne-Marie Girard-Pohjanpelto, Mark Dasenko, Dr. Brent Kronmiller, and Matthew Peterson at the Center of Genome Research and Bioinformatics at Oregon State University (OSU) for their assistance with RNA-sequencing. We thank Drs. Russ Turner and Urszula Iwaniec at the School of Biological and Population Health Sciences at OSU for use of the Lunar PIXImus 2 Dual Energy X-ray Absorptiometer (DXA) instrument. The National Institutes of Health (NIH grants 5R01AT009168 to AFG, CSM, and JFS and 1S10RR027878 to JFS), the Linus Pauling Institute (LPI), the OSU College of Pharmacy, Hopsteiner, Inc, New York, and the OSU Foundation Buhler-Wang Research Fund supported this research. The Marion T Tsefalas Graduate Fellowship from the LPI, the ZRT Laboratory Fund for the LPI, and the Charley Helen, Nutrition Science and Margy J Woodburn Fellowships from the School of Biological and Population Health Sciences at OSU supported YZ.

## Additional information

### Funding

| Funder | Grant reference number | Author |
|---|---|---|
| National Institutes of Health | 5R01AT009168 | Claudia S Maier<br>Jan F Stevens<br>Adrian F Gombart |
| National Institutes of Health | 1S10RR027878 | Jan F Stevens |
| OSU Foundation | Buhler-Wang Research Fund | Cristobal L Miranda |
| Linus Pauling Institute | Marion T. Tsefalas Graduate Fellowship | Yang Zhang |
| Linus Pauling Institute | ZRT Laboratory Fund | Yang Zhang |
| Hopsteiner, Inc New York, NY | | Jan F Stevens |
| OSU School of Biological & Population Health Sciences | | Yang Zhang |

The funders had no role in study design, data collection and interpretation, or the decision to submit the work for publication.

## Author contributions
Yang Zhang, Conceptualization, Resources, Data curation, Formal analysis, Supervision, Funding acquisition, Validation, Investigation, Visualization, Methodology, Writing - original draft, Project administration, Writing - review and editing; Gerd Bobe, Cristobal L Miranda, Conceptualization, Data curation, Formal analysis, Validation, Investigation, Visualization, Methodology, Writing - original draft, Project administration, Writing - review and editing; Malcolm B Lowry, Investigation, Methodology, Writing - review and editing; Victor L Hsu, Conceptualization, Formal analysis, Investigation, Visualization, Methodology, Writing - original draft, Writing - review and editing; Christiane V Lohr, Formal analysis, Investigation, Methodology, Writing - review and editing; Carmen P Wong, Formal analysis, Investigation, Visualization, Methodology, Writing - original draft, Writing - review and editing; Donald B Jump, Resources, Formal analysis, Investigation, Visualization, Methodology, Writing - review and editing; Matthew M Robinson, Validation, Investigation, Visualization, Methodology, Writing - review and editing; Thomas J Sharpton, Resources, Visualization, Methodology, Writing - review and editing; Claudia S Maier, Conceptualization, Resources, Funding acquisition, Validation, Visualization, Methodology, Project administration, Writing - review and editing; Jan F Stevens, Conceptualization, Resources, Funding acquisition, Methodology, Project administration, Writing - review and editing; Adrian F Gombart, Conceptualization, Resources, Formal analysis, Supervision, Funding acquisition, Investigation, Visualization, Methodology, Writing - original draft, Project administration, Writing - review and editing

## Author ORCIDs
Yang Zhang ![ORCID] https://orcid.org/0000-0002-0434-931X
Adrian F Gombart ![ORCID] https://orcid.org/0000-0001-7830-0693

## Ethics
Animal experimentation: This study was performed in strict accordance with the recommendations in the Guide for the Care and Use of Laboratory Animals of the National Institutes of Health. All of the animals were handled according to approved institutional animal care and use committee (IACUC) protocols (ACUP 5053) of Oregon State University.

## Decision letter and Author response
Decision letter https://doi.org/10.7554/eLife.66398.sa1
Author response https://doi.org/10.7554/eLife.66398.sa2

# Additional files

## Supplementary files
• Source data 1. Source datas of *Table 1*, *2*, *3*, *4*.
• Transparent reporting form

## Data availability
RNA-seq data has been deposited in GEO under accession code GSE164636. All data generated or analyzed during this study are included in the manuscript and supporting files. Source data files are provided for Figures 2 - 13 and Tables 1 and 2. To review liver histology images go to: https://doi.org/10.6084/m9.figshare.13619273.

The following datasets were generated:

| Author(s) | Year | Dataset title | Dataset URL | Database and Identifier |
|---|---|---|---|---|
| Zhang Y, Gombart | 2021 | TXN, a Xanthohumol | https://www.ncbi.nlm. | NCBI Gene Expression |

| AF | | | Derivative, Significantly Attenuates High-Fat Diet Induced Hepatic Steatosis In Vivo by Antagonizing PPARγ | nih.gov/geo/query/acc.cgi?acc=GSE164636 | Omnibus, GSE164636 |
|---|---|---|---|---|---|
| Zhang Y, Gombart AF | 2021 | | figshare links for Figures 6, 11 and 1210.6084/m9.figshare.13619273 | Figshare, 10.6084/m9.figshare.13619273 | |

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
