## [Decision Letter]

**Acceptance summary:**

All reviewers found this study interesting and also important. It is potentially of broad interest to the readers of *eLife* and also a valuable addition to the field of metabolism and hepatic steatosis. It provides a comprehensive analysis of a new compound that shows promising therapeutic potential in improving hepatic steatosis and glucose handling in response to metabolic stress.

**Decision letter after peer review:**

Thank you for submitting your article "TXN, a xanthohumol derivative, attenuates high-fat diet induced hepatic steatosis by antagonizing PPARγ" for consideration by eLife. Your article has been reviewed by 3 peer reviewers, and the evaluation has been overseen by a Reviewing Editor and Mone Zaidi as the Senior Editor. The following individual involved in review of your submission has agreed to reveal their identity: José Cesar Rosa Neto (Reviewer #2).

Essential revisions:

1. Authors should assess in more detail the effects on hepatic inflammation and fibrosis (gene and protein markers).

2. Discussion needs to be updated to expand on the following: [1] other potential mechanisms of TXN, [2] the effects of TXN on microbiome and bile acid metabolism, particularly in relation to NAFLD, [3] potential effects of crosstalk between adipose and liver crosstalk in response to TXN treatment, [4] potential sex differences, [5] expand on the balance between PPAR α and γ signaling in the regulation of hepatic steatosis and how TXN may disrupt that balance to improve hepatic lipid profiles.

3. More assays to characterize physical activity in response to TXN treatment (e.g. voluntary running wheel, time to exhaustion on treadmill).

4. Assess lipid oxidation pathway in response to TXN treatment.

5. Assess insulin sensitivity and glucose handling in TXN treated mice.

*Reviewer #1 (Recommendations for the authors):*

In this study, Zhang et al., systematically analyze the effect of xanthohumol and TXN, a xanthohumol derivative, in a model of high-fat diet (HFD) feeding to mice, inducing several pathologies related to the metabolic syndrome. They analyze serum, different fat compartments and the liver. Furthermore, they assess energy expenditure. They convincingly show that these compounds attenuate HFD-induced weight, hepatic steatosis and lipid accumulation in adipose tissues. Furthermore, they newly showed that XN and TXN bind to the PPARγ ligand-binding domain pocket and that this at least in part is responsible for the observe beneficial effects.

This is a comprehensive analysis of XN and TXN effects in different tissue compartments and on different pathological mechanism, respectively. Data are well presented and overall well discussed. Still, there are some issues that should described in more detail.

– Authors should assess in more detail the effects on hepatic inflammation and fibrosis. Even if the (duration of feeding the) diet may not induce histologically detectable changes, markers of hepatic inflammation and fibrosis, including activation of hepatic stellate cells, should be assessed.

– Authors briefly discuss and cite that there are other/further mechanism of action/previous studies by which xanthohumol exhibits beneficial effects in models of diet induced obesity and NAFLD. These should be discussed in more detail, and eventually, it should be also experimentally addressed, whether some of these mechanisms are also operative in the present study.

– It is known that changes in the microbiome and also (associated effects on) bile acid metabolism significantly affect diet induced weight gain and (progression of) NAFLD. Authors should discuss known effects of xanthohumol in this context.

– Also the crosstalk between adipose tissue and (fatty) liver plays a critical role in weight gain as well as NAFLD-progression. Authors should discuss, how xanthohumol may interfere with this crosstalk and also investigate some known key factors such as adipokines, lipocalin, IL6 in their study.

– An intriguing finding of the present study is that XN and TXN act as antagonists of PPARγ. In part, it has already discussed by the authors, but it should be done in more detail: what are the potential clinical implications in NAFLD, diabetes, obesity. When during the course of disease might XN or TXN be applied? Alone or in combination with other PPARγ affecting drugs? What are the risks and potential side effects?

– Authors might also discuss known as well as potential gender-specific differences in the mode of action.

– Regarding synthesis/source of the compounds, authors should not only refer to a previous publication but provide more details in the method section.

*Reviewer #2 (Recommendations for the authors):*

The manuscript titled: " TXN, a Xanthohumol Derivative, Significantly Attenuates High-Fat Diet Induced Hepatic Steatosisin vivo by Antagonizing PPARγ" conduced by Zhang et al., showed that Xanthohumol Derivative reduced the hepatic steatosis by assumption that it is the PPAR-γ dependent mechanism.

I suggest that the manuscript should be rejected. The *eLife* is an outstanding journal and it is necessary high criticism on the evaluation of the manuscript. The conclusion is not supported by the method, it is necessary many additional experiments to show the relevance effect of TXN on lipid metabolism on liver.

The Results section show discussion during the description of results, for instance phrase that starts on line 188. On the same direction, the paragraph that starts on line 293.

The discussion is very speculative, and the authors starts the discussion writing about the AMPK and inflammation but neither of these parameters were evaluated during the manuscript.

Weakness of the manuscript:

The authors suggest that the effects of TXN might be associated by increase on physical activity. But the parameters used to measure are very weak. It is necessary to show the increased on physical activity by evaluation of running on wheel voluntary running cage. Moreover, the authors might measure if the TXN is able to improve the time to exhaustion in treadmill running.

The steatosis is a very complex process, which can induced by huge lipid storage or reduction on lipid oxidation. The unbalance between synthesis and oxidation is crucial to understand the hepatic steatosis. The authors did not measure the enzymes that regulate the lipid oxidation on liver, in special the PPAR-α pathway.

Moreover, Souza CO et al., 2020 showed (1) that the PPAR-γ deletion on myeloid cell, induces the stronger insulin resistance. On the same direction the ablation of PPAR-γ on myeloid cell induces the huge pro-inflammatory response (2). The beneficial effects of ppar-γ antagonism in hepatic steatosis are still unclear. The others authors showed that the a treatment with PPAR-γ agonist is able to improve the NAFLD and NASH induced by HFD (3). The dual role of PPAR-γ is showed with the deletion of PPAR-γ in hepatocytes that mitigate the NAFLD in mice fed with HFD, but this phenotype shows metabolic complications, such as inflammation and disturb on glucose homeostasis (4).

Another question is about the adipose tissue homeostasis. The great problem of obesity is the increase of fatty acids storage in non-adipose tissue. The effects of antagonist TXN is able to impair the metabolic status of adipose tissue. Finally, the insulin resistance is the first step of lipids accumulation on the hepatic parenchyma and it is not evaluate in this study (5).

Thus, my conclusion is that several steps are unclear in this study and it is possible to lead a misinterpretation.

*Reviewer #3 (Recommendations for the authors):*

Overall, this study is comprehensive in its findings and provides valuable insights into the potential use of natural XN and synthetic TXN as novel and low-cost therapeutics for diet-induced hepatic steatosis. We suggest the following items be addressed to improve the clarity of the work:

1. In the title of the manuscript, it would be preferable to indicate full name of compound Tetrahydroxanthohumol.

2. Line 60 – specify adipocyte hypertrophy.

3. Doses are presented as "daily oral intake of TXN at 30 mg/kg body weight (BW)" and "XN at a daily dose of 30 mg/kg or 60 mg/kg BW". Based on Table 3, the TXN and XN are present at 30 or 60 mg of compound per kg of diet; therefore, the dose given per body weight must be an error that needs to be corrected in main text and Table 3. The XN and TXN compounds are formulated with the diet, therefore the daily dose of compound consumed by each mouse will vary depending on the quantity of food consumed and the body weight of the animal, which increases over the period of the study. It would be preferable to indicate the percentage of compound in the diet, which is 0.003% TXN (Table 3, 0.003 g /100 g diet) and 0.003% or 0.006% XN (Table 3, 0.003 g or 0.006 g/100 g diet). The dose of compound per mouse over time could be estimated from body weights and average daily food intake data in the phenome_feeding.csv file.

4. Line 97 and 99 – Hyphens are not necessary in certain contexts (e.g. "TXN-supplementation") and not being used consistently throughout manuscript, (e.g. "XN supplementation") therefore please consider removing all such instances.

5. In Lines 112 – 114 below, it is not clear whether the decreased food intake being referred to is a comparison within the HXN group (i.e. compared to week 0) or compared to the LFD group at the indicated weeks. Please clarify.

"HXN-treated mice adapted better to the HFD, indicated by decreased food intake at week 1, 6-10, 13, and 16 (p < 0.05), and caloric intake (p = 0.01) (Figure 2C-D), resulting in less BW gain."

6. Figure 3, Supplement 2: It is unclear why the r and p values reported for this figure in the manuscript (e.g. Lines 217 – 222) are different from what is reported in the figure, please see excerpt below. Were different correlation tests used?

Namely, there was an inverse relationship between caloric intake and plasma TAG among LFD mice (Spearman, r = -0.60, p = 219 0.04; Figure 3—figure supplement 2 A1), which was lost on the HFD (Spearman, r = 0.12, p = 0.70; Figure 3—figure supplement 2 A2). TXN treatment restored the negative correlation between caloric intake and plasma TAG (Spearman r = -0.65, p = 0.04; Figure 3—figure supplement 2 A5).

7. Line 215, define triglyceride (TAG)

8. Line 238, energy expenditure was indicated in Figure 3C not 4C

"In contrast to energy expenditure (Figure 3C)…"

9. Line 291: change three to "3-fold vs. 2.5-fold increase…

10. Lines 289 – 292 describe data in three panels, Figure 5A – C (not just Figure 5 A, C)

11. In Figure 5, the middle column of data should be labeled B1 – B5 and last columns of data should be labeled C1 – C5. Also, abbreviations used are found in text but not in Figure 5 legend therefore please add following labeling to Figure 5 for convenience of reader:

– after A. Subcutaneous fat, please add (sWAT)

– after B. Epididymal fat, please add (eWAT)

– after C. Mesenteric fat, please add (mWAT)

12. For clarity, please specify p = 0.06 in Figure 5B. Legend indicates that p< 0.05 was considered significant, therefore please consider changing wording in Lines 293- 296, to indicate that TXN-treated mice trended higher than HFD group:

"Compared to the HFD group, a smaller but significant increase in eWAT adipose tissue weight was observed in HXN-treated mice while that of TXN-treated mice trended higher (p = 0.06) (Figure 5B)."

13. Define abbreviation in Line 336: low density lipoprotein receptor knock-out (LDLR-/-).

14. Consider restructuring the sentence below for simpler comprehension and indicate Figure 6A at end of sentence.

Change lines 338-340: "Supplementation with XN decreased in a dose-dependent manner the number and size of intrahepatic lipid vacuoles in HFD mice".

"XN supplementation decreased the number and size of intrahepatic lipid vacuoles in HFD mice in a dose-dependent manner (Figure 6A)."

15. Line 341: At end of sentence please specify (Figure 6A).

16. Line 347: At end of sentence please specify (Figure 6B).

17. Please consider below the reorganized description of Figure 3B results (lines 353- 358) for better clarity:

Both HXN and TXN supplementation decreased liver lipid accumulation on a HFD by two-fold (Figure 3B). Three of 12 HXN-supplemented mice and 5 of 11 TXN-supplemented mice had less than 5% lipid area while 7 of 12 HXN-supplemented mice 9 of 11 TXN-supplemented mice had less than 10% lipid area (Figure 3B). In comparison, only 1 of 12 HFD control mice were below 10% lipid area (Figure 3B).

18. Line 421 please define abbreviation: Furthermore, Kyoto Encyclopedia of Genes and Genomes (KEGG) pathway…

19. Line 458 please define abbreviation: We implemented support vector machine (SVM).

20. Missing the word "are" at beginning of Line 469: are known target genes of…

21. Line 507: PPARγ has two major isoforms, γ1 and γ2, generated from the same gene by alternative splicing, so it is unclear why Pparγ2 is indicated as "a predicted PPAR target gene". There is no description of relevance of Plin4 gene, which is presented in Figure 10, and Figure 10 legend does not mention Plin4. Please address.

22. Results section for Figures 11 and 12 seem to be scrambled. To improve clarity, please consider making the following rearrangements and edits to lines 537 – 547:

"We tested escalating doses of XN and TXN (Strathmann and Gerhauser, 2012). After treatments, we determined the number of live cells using an MTT assay. XN and TXN were only significantly cytotoxic for 3T3-L1 cells at a dose of 50 μM (data not shown). While it is difficult to translate in vivo doses to in vitro doses, based on previous in vitro studies (Yang et al., 2007; Samuels, Shashidharamurthy and Rayalam, 2018) and our current cell viability data, we selected low (5 μM), medium (10 μM) and high (25 μM) concentrations of XN and TXN for the subsequent experiments where cell viability was greater than 90% (data not shown). Murine preadipocyte 3T3-L1 differentiation and …"

Also, the sentence shown below, presently Lines 538 – 540 seems to be information pertaining to Figure 12 and should be moved to section 8:

3T3-L1 cells were treated with 0.1% DMSO, 1 μM rosiglitazone (RGZ), 1 μM GW9662, XN (5, 10 and 25 μM), TXN (5, 10 and 25 μM), 25 μM XN + 1 μM RGZ or 25 μM TXN + 1 μM RGZ for 48 h.

---

## [Author Response]

1. Authors should assess in more detail the effects on hepatic inflammation and fibrosis (gene and protein markers).

We extracted genes involved in both hepatic fibrosis and inflammation from source file other Data DEGs_LV.xlsx and put the expression data in Tables 1 and 2, respectively. We did not observe transcriptional regulation of liver mRNAs that encode proteins involved in these pathways with HFD feeding. We also performed Sirius red staining for the deposition of collagen fibers to determine the occurrence of fibrosis and did not observe staining in any of the mice.

Inserted text line 427, in yellow highlight: We did not detect discernible fibrosis in liver sections using Sirius red staining in any of the mice (data not shown).

Inserted text line 549, in yellow highlight: Consistent with the lack of Sirius red staining in the liver, we observed no changes in expression of genes involved in hepatic fibrosis in the HFD mice compared with the LFD. In response to TXN treatment, we noted a 4-fold decrease in *Timp2* and *Col1a1* both factors that promote hepatic fibrosis (Table 1) (Nie et al., 2004; Chakraborty, Oakley and Walsh, 2012).

Inserted text line 559, in yellow highlight: We then examined transcript levels for genes in pathways regulated by PPARα, namely lipid oxidation. We observed no change with TXN treatment (Table 4). Consistent with the GO enrichment analysis, most of the changes were for genes encoding proteins involved in the lipid storage pathway (Table 4) and regulated by PPARγ.

2. Discussion needs to be updated to expand on the following:i. Other potential mechanisms of TXN,

Author response: To expand on other potential mechanisms we added the following additional text inserted at line 975-1011 (in yellow highlight, see text below) to discuss other potential mechanisms.

As we discussed earlier, antagonizing PPARγ action is likely an additional mechanism by which XN and TXN suppress diet-induced NAFLD. Other possible mechanisms may play a role as well. Gut microbiota dysbiosis associates with obesity and T2D, diseases strongly associated with NAFLD. Mouse studies and fecal transplantation experiments have demonstrated a causal role of gut microbiota in the development of NAFLD (Henao-Mejia et al., 2012). We previously reported both XN and TXN drastically change gut microbiota composition in C57Bl6/J male mice, accompanied by a significant change in fecal bile acid composition (Zhang et al., 2020). Specifically, administration of XN and TXN decreased intestinal microbiota diversity and abundance, altered bile acid metabolism, and reduced inflammation. Changes in gut microbiota and bile acid metabolism may also explain, in part, the improvements in MetS and NAFLD, but requires further investigation in future studies.

Legette et al. reported feeding XN to Zucker fa/fa rats for six weeks significantly lowered BW gain and plasma glucose levels only in male but not female rats (Legette et al., 2013). This gender difference in response to XN is not unique as similar findings were observed for other flavonoids (Camper-Kirby Dreama et al., 2001; Blair et al., 2002; Guo et al., 2005). Consistent with a prior study (Zhou, Jiang and Rui, 2009) we found expression of numerous major urinary protein (*Mup*) genes reduced by a HFD as compared to the LFD; however, the expression of these same genes were induced by TXN administration (Table 3). MUPs are unique members of the lipocalin super-family produced by the liver and secreted into urine primarily in males (Zhou and Rui, 2010). They function in the urine as pheromones in chemical communication and as metabolic signals regulating glucose and lipid metabolism in individual animal (Zhou and Rui, 2010). Because adult male mice secrete significantly more MUPs than females, this finding may explain why female mice do not respond to XN and TXN like their male counterparts. TXN treatment also beneficially modulated expression of lipocalin members, Lcn2 and ApoM (Wang et al., 2007; Milner et al., 2009; Auguet et al., 2013; Yang et al., 2019). To elucidate the role these changes in gene expression play in gender-specific responses to TXN will require additional research.

To maintain energy homeostasis, proper crosstalk between metabolically active tissues is essential (Stern, Rutkowski and Scherer, 2016). In NAFLD (and MetS in general), these tissues often present a chronic low-grade inflammation characterized by the recruitment of proinflammatory cells, cytokines, and acute-phase proteins (Lackey and Olefsky, 2016; Wang et al., 2021). Previously we reported that supplementation with TXN decreased chronic inflammation with reduced expression of major proinflammatory cytokines *Il-6* and *Tnfα* in WAT and to a smaller extent in the liver (Zhang et al., 2020). We also observed a decrease in *Ccl2*, a chemotactic factor involved in the recruitment of monocytes, and macrophage marker F4/80 in WAT suggesting that TXN may protect WAT from macrophage infiltration (Zhang et al., 2020).

ii. The effects of TXN on microbiome and bile acid metabolism, particularly in relation to NAFLD,

Please see text above and inserted at lines 975-985 in manuscript.

iii. Potential effects of crosstalk between adipose and liver crosstalk in response to TXN treatment,

Please see text above and inserted at lines 1003-1011 in the manuscript.

iv. Potential sex differences,

Please see text above and inserted at lines 986-1002 in the manuscript.

v. Expand on the balance between PPAR α and γ signaling in the regulation of hepatic steatosis and how TXN may disrupt that balance to improve hepatic lipid profiles.

We did not observe changes in expression of genes encoding proteins involved in processes like lipid oxidation regulated by the PPARα pathway (Table 4).

We inserted text at lines 559-562 to describe, We then examined transcript levels for genes in pathways regulated by PPARα, namely lipid oxidation. We observed no change with TXN treatment (Table 4). Consistent with the GO enrichment analysis, most of the changes were for genes encoding proteins involved in the lipid storage pathway (Table 4) and regulated by PPARγ.

3. More assays to characterize physical activity in response to TXN treatment (e.g. voluntary running wheel, time to exhaustion on treadmill).

While the comment posed by the Reviewer is important, in this paper, the technology present in our metabolic cages limited the type of energy expenditure data we could collect as the cages lacked running wheels. To address the Reviewer’s request would require purchasing the additional equipment and repeating the experiment. This is not feasible, as funding for this project has ended. We need to acquire additional funding for future work in which we would design a study to better address the physical activity question posed by the reviewer.

4. Assess lipid oxidation pathway in response to TXN treatment.

We did not observe regulation of genes involved in lipid oxidation (Table 4).

5. Assess insulin sensitivity and glucose handling in TXN treated mice.

GTT and fasting insulin, fasting glucose and HOMA-IR data is incorporated into the main text (line 120-135), Materials and methods section (line 1075-1082), see Figure 2-supplement 1.

To measure the effect of XN and derivatives on glucose homeostasis, we performed a glucose tolerance test (GTT) after feeding the corresponding diets for 9 weeks. GTT results showed impaired glucose clearance in HFD control mice (Figure 2-supplement 1 A, dashed blue line; Figure 2-supplement 1 B). Compared to HFD control mice, TXN treated mice showed significantly improved glucose clearance, as indicated at time points 30 min, 60 min and 120 min post i.p. injection (Figure 2-supplement 1 A, green line; p-values = 0.04, 0.02, and < 0.01, respectively), and a significantly lower AUC (area under the curve) (Figure 2-supplement 1 B, p < 0.01). HXN treated mice also showed improved glucose clearance at time points 60 min and 120 min post i.p. injection (Figure 2-supplement 1 A, red line; p-values = 0.04 and 0.05, respectively). Although not statistically significant, HXN treated mice showed a trend toward a lower AUC (Figure 2-supplement 1 B, p = 0.067). LXN treatment did not improve glucose clearance (Figure 2-supplement 1 A, orange line; Figure 2-supplement 1 B).

While fasting glucose was not different between TXN-treated and HFD control mice after 16 weeks of feeding (Figure 2-supplement 1 C, p = 0.56), fasting insulin was significantly improved by TXN treatment as suggested by lower circulating insulin (Figure 2-supplement 1 D, p = 0.003) and HOMA-IR (Figure 2-supplement 1 E, p = 0.001). Consistent with our previously published study (Miranda et al., 2018), these results indicate that TXN significantly improved glucose homeostasis; XN seems to have a dose response as HXN appears to be more effective than LXN.

We have also prepared an Evaluation Summary and Public Reviews of your work below, which are designed to transform your manuscript into a preprint with peer reviews.Reviewer #1 (Recommendations for the authors):In this study, Zhang et al., systematically analyze the effect of xanthohumol and TXN, a xanthohumol derivative, in a model of high-fat diet (HFD) feeding to mice, inducing several pathologies related to the metabolic syndrome. They analyze serum, different fat compartments and the liver. Furthermore, they assess energy expenditure. They convincingly show that these compounds attenuate HFD-induced weight, hepatic steatosis and lipid accumulation in adipose tissues. Furthermore, they newly showed that XN and TXN bind to the PPARγ ligand-binding domain pocket and that this at least in part is responsible for the observe beneficial effects.This is a comprehensive analysis of XN and TXN effects in different tissue compartments and on different pathological mechanism, respectively. Data are well presented and overall well discussed. Still, there are some issues that should described in more detail.

We thank the Reviewer for their time and effort in reviewing the manuscript. We appreciate the Reviewer’s positive feedback regarding the study and will address the issues they have highlighted.

– Authors should assess in more detail the effects on hepatic inflammation and fibrosis. Even if the (duration of feeding the) diet may not induce histologically detectable changes, markers of hepatic inflammation and fibrosis, including activation of hepatic stellate cells, should be assessed.

We extracted genes involved in both hepatic fibrosis and inflammation from source file otherData DEGs_LV.xlsx and subsequently put in Tables 1 and 2, respectively. We did not observe transcriptional regulation of liver mRNAs that encode proteins involved in these pathways with HFD feeding. We also performed Sirius red staining for the deposition of collagen fibers to determine the occurrence of fibrosis and did not observe staining in any of the mice.

Inserted text at line 553, We also did not observe changes in expression for transforming growth factor (*Tgfb*) or platelet-derived growth factor (*Pdgf*), key factors in driving hepatic stellate cell activation following hepatocellular injury (data not shown) (Dooley et al., 2001; Tsuchida et al., 2017).

Inserted text line 427, in yellow highlight: We did not detect discernable fibrosis in liver sections using Sirius red staining in any of the mice (data not shown).

Inserted text line 549, in yellow highlight: Consistent with the lack of Sirius red staining in the liver, we observed no changes in expression of genes involved in hepatic fibrosis in the HFD mice compared with the LFD (Table 1). In response to TXN treatment, we noted a 4-fold decrease in *Timp2* and *Col1a1* both factors that promote hepatic fibrosis (Table 1) (Nie et al., 2004; Chakraborty, Oakley and Walsh, 2012).

Inserted text line 556 regarding inflammation, Finally, we did not observe increased expression of genes involved in inflammation with 16-weeks of HFD feeding, but did observe a significant decrease in *Ccr2* and *Fgf21* expression with TXN treatment (Table 1).

– Authors briefly discuss and cite that there are other/further mechanism of action/previous studies by which xanthohumol exhibits beneficial effects in models of diet induced obesity and NAFLD. These should be discussed in more detail, and eventually, it should be also experimentally addressed, whether some of these mechanisms are also operative in the present study.

We have inserted extensive text (line 975 to 1011) in response to this comment in the discussion. We agree addressing whether some of these mechanisms are operative is important; however, we envision these additional experiments in future work.

As we discussed earlier, antagonizing PPARγ action is likely an additional mechanism by which XN and TXN suppress diet-induced NAFLD. Other possible mechanisms may play a role as well. Gut microbiota dysbiosis associates with obesity and T2D, diseases strongly associated with NAFLD. Mouse studies and fecal transplantation experiments have demonstrated a causal role of gut microbiota in the development of NAFLD (Henao-Mejia et al., 2012). We previously reported both XN and TXN drastically change gut microbiota composition in C57Bl6/J male mice, accompanied by a significant change in fecal bile acid composition (Zhang et al., 2020). Specifically, administration of XN and TXN decreased intestinal microbiota diversity and abundance, altered bile acid metabolism, and reduced inflammation. Changes in gut microbiota and bile acid metabolism may also explain, in part, the improvements in MetS and NAFLD, but requires further investigation in future studies.

Legette et al., reported feeding XN to Zucker fa/fa rats for six weeks significantly lowered BW gain and plasma glucose levels only in male but not female rats (Legette et al., 2013). This gender difference in response to XN is not unique as similar findings were observed for other flavonoids (Camper-Kirby Dreama et al., 2001; Blair et al., 2002; Guo et al., 2005). Consistent with a prior study (Zhou, Jiang and Rui, 2009) we found expression of numerous major urinary protein (*Mup*) genes reduced by a HFD as compared to the LFD; however, the expression of these same genes were induced by TXN administration (Table 3). MUPs are unique members of the lipocalin super-family produced by the liver and secreted into urine primarily in males (Zhou and Rui, 2010). They function in the urine as pheromones in chemical communication and as metabolic signals regulating glucose and lipid metabolism in individual animal (Zhou and Rui, 2010). Because adult male mice secrete significantly more MUPs than females, this finding may explain why female mice do not respond to XN and TXN like their male counterparts. TXN treatment also beneficially modulated expression of lipocalin members, Lcn2 and ApoM (Wang et al., 2007; Milner et al., 2009; Auguet et al., 2013; Yang et al., 2019). To elucidate the role these changes in gene expression play in gender-specific responses to TXN will require additional research.

To maintain energy homeostasis, proper crosstalk between metabolically active tissues is essential (Stern, Rutkowski and Scherer, 2016). In NAFLD (and MetS in general), these tissues often present a chronic low-grade inflammation characterized by the recruitment of proinflammatory cells, cytokines, and acute-phase proteins (Lackey and Olefsky, 2016; Wang et al., 2021). Previously we reported that supplementation with TXN decreased chronic inflammation with reduced expression of major proinflammatory cytokines *Il-6* and *Tnfα* in WAT and to a smaller extent in the liver (Zhang et al., 2020). We also observed a decrease in *Ccl2*, a chemotactic factor involved in the recruitment of monocytes, and macrophage marker F4/80 in WAT suggesting that TXN may protect WAT from macrophage infiltration (Zhang et al., 2020).

– It is known that changes in the microbiome and also (associated effects on) bile acid metabolism significantly affect diet induced weight gain and (progression of) NAFLD. Authors should discuss known effects of xanthohumol in this context.

The Reviewer makes a valid point and we have added additional text to the Discussion to describe the known effects of xanthohumol in this context.

Inserted text, lines 975-985: As we discussed earlier, antagonizing PPARγ action is likely an additional mechanism by which XN and TXN suppress diet-induced NAFLD. Other possible mechanisms may play a role as well. Gut microbiota dysbiosis associates with obesity and T2D, diseases strongly associated with NAFLD. Mouse studies and fecal transplantation experiments have demonstrated a causal role of gut microbiota in the development of NAFLD (Henao-Mejia et al., 2012). We previously reported both XN and TXN drastically change gut microbiota composition in C57Bl6/J male mice, accompanied by a significant change in fecal bile acid composition (Zhang et al., 2020). Specifically, administration of XN and TXN decreased intestinal microbiota diversity and abundance, altered bile acid metabolism, and reduced inflammation. Changes in gut microbiota and bile acid metabolism may also explain, in part, the improvements in MetS and NAFLD, but requires further investigation in future studies.

– Also the crosstalk between adipose tissue and (fatty) liver plays a critical role in weight gain as well as NAFLD-progression. Authors should discuss, how xanthohumol may interfere with this crosstalk and also investigate some known key factors such as adipokines, lipocalin, IL6 in their study.

The Reviewer makes an important point and we have added additional text to the Discussion to describe the known effects of xanthohumol in this context.

Inserted text, lines 1003-1011: To maintain energy homeostasis, proper crosstalk between metabolically active tissues is essential (Stern, Rutkowski and Scherer, 2016). In NAFLD (and MetS in general), these tissues often present a chronic low-grade inflammation characterized by the recruitment of proinflammatory cells, cytokines, and acute-phase proteins (Lackey and Olefsky, 2016; Wang et al., 2021). Previously we reported that supplementation with TXN decreased chronic inflammation with reduced expression of major proinflammatory cytokines *Il-6* and *Tnfα* in WAT and to a smaller extent in the liver (Zhang et al., 2020). We also observed a decrease in *Ccl2*, a chemotactic factor involved in the recruitment of monocytes, and macrophage marker F4/80 in WAT suggesting that TXN may protect WAT from macrophage infiltration (Zhang et al., 2020).

– An intriguing finding of the present study is that XN and TXN act as antagonists of PPARγ. In part, it has already discussed by the authors, but it should be done in more detail: what are the potential clinical implications in NAFLD, diabetes, obesity. When during the course of disease might XN or TXN be applied? Alone or in combination with other PPARγ affecting drugs? What are the risks and potential side effects?

While the Reviewer raises important questions, it is beyond the scope of this paper to answer all of these questions without extensive speculation. We would need to conduct future studies investigating the efficacy of TXN on reversing established disease in obese animals. We are interested in the question of when during the course of disease might these compounds work and are there effective combinations with other PPARγ ligands. We also realize it is important to perform dose escalation studies with TXN to determine if it has potential side effects or risks in preclinical models. At the dose used in this study, we do not observe side effects. We do know side effects in mice and rats are not reported with XN at much higher doses than those used in this study, but we have not established this for TXN. We also do not know if we can use lower doses of TXN and still see beneficial effects. Pharmacokinetic studies in animal models such as those done with XN are also warranted before considering clinical studies in humans. We have added text to the final paragraph in the discussion address these points.

We rewrote the text, lines 1012-1030: In conclusion, we demonstrated the dose of TXN given in the diet is very effective in suppressing the development and progression of diet induced hepatic steatosis in mice. TXN appears more effective in vivo than XN perhaps due to significantly higher levels of TXN in the liver, but XN can slow progression of the condition at a higher dose. At the dose used for TXN, we have not observed adverse events in our prior or current studies (Miranda et al., 2018). In future work, we would consider testing lower and higher doses for safety and efficacy and pursuing pharmacokinetic studies such as those already done with XN (Legette et al., 2012; Legette et al., 2014). While XN and TXN are effective preventative approaches in rodents, in future studies we are interested in determining if these compounds can treat existing obesity. We provide evidence that XN and TXN act as novel, natural and synthetic antagonists of PPARγ that bind with a similar affinity as the agonist PGZ. Our findings support further development of XN and TXN as novel, low-cost therapeutic compounds for diet-linked hepatic steatosis with fewer negative side effects than current drugs (e.g., reduced adipose tissue expansion). Additionally, the structures of XN and TXN could serve as scaffolds for the synthesis of more effective compounds to treat NAFLD and other fatty liver diseases. These findings also raise the possibility of testing XN and TXN in combination with other PPARγ ligands in treating obesity and metabolic syndrome. Although these results are encouraging, further studies are required to clarify possible use in humans for the prevention and treatment of diet-linked hepatic steatosis.

– Authors might also discuss known as well as potential gender-specific differences in the mode of action.

While we do not fully understand the reasons for the gender-specific differences, we have added additional text to the Results and Discussion to describe the known effects of xanthohumol in this context and propose possible explanations.

In the Results line 542, we included the text genes from the major urinary protein family (Table 3), to point out that these genes were upregulated by TXN treatment.

Inserted text, lines 986-1002: Legette et al. reported feeding XN to Zucker fa/fa rats for six weeks significantly lowered BW gain and plasma glucose levels only in male but not female rats (Legette et al., 2013). This gender difference in response to XN is not unique as similar findings were observed for other flavonoids (Camper-Kirby Dreama et al., 2001; Blair et al., 2002; Guo et al., 2005). Consistent with a prior study (Zhou, Jiang and Rui, 2009) we found expression of numerous major urinary protein (*Mup*) genes reduced by a HFD as compared to the LFD; however, the expression of these same genes were induced by TXN administration (Table 3). MUPs are unique members of the lipocalin super-family produced by the liver and secreted into urine primarily in males (Zhou and Rui, 2010). They function in the urine as pheromones in chemical communication and as metabolic signals regulating glucose and lipid metabolism in individual animal (Zhou and Rui, 2010). Because adult male mice secrete significantly more MUPs than females, this finding may explain why female mice do not respond to XN and TXN like their male counterparts. TXN treatment also beneficially modulated expression of lipocalin members, Lcn2 and ApoM (Wang et al., 2007; Milner et al., 2009; Auguet et al., 2013; Yang et al., 2019). To elucidate the role these changes in gene expression play in gender-specific responses to TXN will require additional research.

– Regarding synthesis/source of the compounds, authors should not only refer to a previous publication but provide more details in the method section.

To clarify the source of the compounds, we inserted the following text at line 1042 of the Materials and methods.

XN and TXN (both of purity >99%) were provided by Hopsteiner, Inc (New York, NY, USA).

Reviewer #2 (Recommendations for the authors):I suggest that the manuscript should be rejected. The eLife is an outstanding journal and it is necessary high criticism on the evaluation of the manuscript. The conclusion is not supported by the method, it is necessary many additional experiments to show the relevance effect of TXN on lipid metabolism on liver.

The journal *eLife* is an outstanding journal and critical review of our manuscript is justified no matter where we submit it for publication. We will address the comments from this Reviewer and those of the other Reviewers to show the relevance of our findings to the current knowledge in this field. We thank the Reviewer for their time and effort in reviewing the manuscript.

The Results section show discussion during the description of results, for instance phrase that starts on line 188. On the same direction, the paragraph that starts on line 293.

We had trouble locating the specific instances raised by the Reviewer, as lines 188 and 293 were sentences describing the data source. We looked through the entire result sections for text resembling discussion. We removed lines 268-274. The following sentences were removed: In summary, HFD-fed mice used more energy for maintaining basal metabolism, body tissue turnover, or digestion as indicated by a higher energy expenditure and lower directed ambulatory locomotion activity than LFD mice. Compared to HFD only mice, XN- and TXN-treated mice had lower energy expenditure and higher directed ambulatory locomotion and fine movement activities, indicating lower energy for maintaining basal metabolism, body tissue turnover, or digestion and remained more physically active than untreated HFD-fed mice.

The discussion is very speculative, and the authors starts the discussion writing about the AMPK and inflammation but neither of these parameters were evaluated during the manuscript.

We started the discussion by introducing published mechanisms describing how XN and its derivatives ameliorate diet-induced obesity, metabolic perturbations and liver function with the intention of describing the current state of knowledge in the field, not to confirm these mechanisms in our study. Our findings highlight another new, potential mechanism (antagonizing PPARγ) in addition to those already reported. Our point is to discuss what the current paper adds to the field and highlight that plant polyphenols are likely affecting numerous pathways in mediating their effects.

Weakness of the manuscript:The authors suggest that the effects of TXN might be associated by increase on physical activity. But the parameters used to measure are very weak. It is necessary to show the increased on physical activity by evaluation of running on wheel voluntary running cage. Moreover, the authors might measure if the TXN is able to improve the time to exhaustion in treadmill running.

These proposed experiments are good ideas, but not attainable now, as they require us to start a new animal study and to purchase the necessary equipment such as wheels and treadmills that our current funding situation does not allow.

The method we used to measure physical activity in this study is insufficient to address the points described by the reviewer. In this study, we derived physical activity from directed ambulatory locomotion and fine movements collected by the metabolic cage system we used (line # 1092-1107). We did not design the current study to specifically test whether XN and TXN alter physical activity because we did not have evidence that it might. We intentionally did not emphasize this data in the manuscript, because we understood the limitations. We decided to show it because no research groups have reported this phenomenon with xanthohumol and/or its derivatives and it could interest researchers in the field. We agree these findings require designing appropriate studies, as suggested by the Reviewer, for future studies.

The steatosis is a very complex process, which can induced by huge lipid storage or reduction on lipid oxidation. The unbalance between synthesis and oxidation is crucial to understand the hepatic steatosis. The authors did not measure the enzymes that regulate the lipid oxidation on liver, in special the PPAR-α pathway.

We appreciate the Reviewer’s comment and fully agree that steatosis is a complicated process associated with an array of changes in glucose, fatty acid, and lipid metabolism across all tissues. To acquire some insight into this complicated relationship, we used a systematic pathway analysis of liver RNAseq data. We observed no evidence of changes in lipid oxidation by either XN or TXN treatment under our experimental conditions. We also looked for specific genes known in the β-oxidation process and found no differentially expressed genes (DEGs) between the TXN and HFD groups. (Tables 1, 2; line 502 and 513).

On the other hand, we observed many DEGs involved in lipid storage. With limited resources, we decided to focus on lipid synthesis and storage rather than lipid oxidation. With this focus, our data led us to the main finding of this manuscript, which is XN, and TXN can function as PPARγ antagonists. Nevertheless, our findings do not rule out a role for PPARα, but we have not determined if XN and TXN affect PPARα. In future research, we are interested in the possible role of other nuclear receptors including PPARα, but this research is outside the focus of the current manuscript.

Moreover, Souza CO et al., 2020 showed (1) that the PPAR-γ deletion on myeloid cell, induces the stronger insulin resistance. On the same direction the ablation of PPAR-γ on myeloid cell induces the huge pro-inflammatory response (2). The beneficial effects of ppar-γ antagonism in hepatic steatosis are still unclear. The others authors showed that the a treatment with PPAR-γ agonist is able to improve the NAFLD and NASH induced by HFD (3). The dual role of PPAR-γ is showed with the deletion of PPAR-γ in hepatocytes that mitigate the NAFLD in mice fed with HFD, but this phenotype shows metabolic complications, such as inflammation and disturb on glucose homeostasis (4).

There is evidence that PPARγ agonists are beneficial in ameliorating NAFLD and other diet induced metabolic diseases as the Reviewer pointed out. However, it does not contradict the fact that PPARγ antagonists can also be beneficial, as presented in our manuscript and others (Shiomi et al., 2015; Fraunhofer Gesellschaft zur Förderung der Angewandten Forschunge 2017; Ammazzalorso and Amoroso, 2019). The Reviewer cited a study showing PPARγ deletion in macrophages can induce inflammation and deteriorate insulin resistance. However, we did not observe either of these conditions in our study. In contrast, TXN as a PPARγ antagonist did not promote hepatic inflammation (Table 2, line# 513) but improved glucose clearance (Figure 2-supplement 1). We postulate complete absence of PPARγ is quite different from modulating its activity through agonists and antagonists and this may explain the differences noted between the knockout studies and our findings.

We added text to this effect at lines 961-963, Hepatocyte- and macrophage-specific PPARγ deficiency protects *ob/ob* mice from hepatic steatosis (Matsusue et al., 2003; Morán-Salvador et al., 2011);…. and 967-974, Ablation of PPARγ in murine myeloid cells increased insulin resistance (Souza et al., 2020) and ablation in macrophages and hepatic stellate cells, but not hepatocytes increased inflammation (Morán-Salvador et al., 2013). Nevertheless, we did not observe either of these conditions in our study. In contrast, TXN did not promote hepatic inflammation (Table 2) but improved glucose clearance (Figure S1). We postulate complete absence of PPARγ is quite different from modulating its activity through agonists and antagonists and this may explain the differences noted in some of these cell-specific knockout studies and our findings.

Moreover, PPARγ agonists, such as TZDs, are possible double-edged swords: TZD treatments improve hepatic steatosis through insulin-sensitizing effects on the adipose tissue (Ratziu et al., 2010). Long-term TZD treatments will promote unwanted weight gain (Fonseca, 2003) due to TZD’s direct action on adipose tissue, as well as cardiovascular complications (Nesto et al., 2004). Our data suggest limited or no direct action of TXN on adipose tissue, given TXN’s level is undetectable in the adipose tissue (data not shown). Indeed, future pharmacokinetics studies are necessary to determine the systemic distribution of TXN in the host.

Another question is about the adipose tissue homeostasis. The great problem of obesity is the increase of fatty acids storage in non-adipose tissue. The effects of antagonist TXN is able to impair the metabolic status of adipose tissue.

We specifically quantified three different fat depots, namely epididymal, subcutaneous, and mesenteric white adipose tissue. Our data showed no sign of TXN impairing the metabolic status of adipose tissue, in fact, we saw the opposite. One explanation could be that TXN is undetectable in the adipose tissue, therefore cannot physically bind to PPARγ and antagonize its actions. In fact, this may explain why TXN is a better PPARγ antagonist compared with traditional PPARγ agonists that cause unintentional/undesirable weight gain in the adipose tissue.

Finally, the insulin resistance is the first step of lipids accumulation on the hepatic parenchyma and it is not evaluate in this study (5).

We have added data from glucose tolerance testing, and fasting glucose and insulin data to the Results section. See Figure 2-supplement 1.

Thus, my conclusion is that several steps are unclear in this study and it is possible to lead a misinterpretation.

While gaps may exist, we try to carefully interpret our data, avoid overstating findings and list caveats where necessary to avoid misinterpretation.

Reviewer #3 (Recommendations for the authors):Overall, this study is comprehensive in its findings and provides valuable insights into the potential use of natural XN and synthetic TXN as novel and low-cost therapeutics for diet-induced hepatic steatosis. We suggest the following items be addressed to improve the clarity of the work:

We thank the Reviewer for their time and effort in reviewing the manuscript. We appreciate the Reviewer’s positive feedback regarding the study and will address the issues they have highlighted to improve clarity.

1. In the title of the manuscript, it would be preferable to indicate full name of compound Tetrahydroxanthohumol.

We included the full name in the title.

2. Line 60 – specify adipocyte hypertrophy.

We specified the term as suggested.

3. Doses are presented as "daily oral intake of TXN at 30 mg/kg body weight (BW)" and "XN at a daily dose of 30 mg/kg or 60 mg/kg BW". Based on Table 3, the TXN and XN are present at 30 or 60 mg of compound per kg of diet; therefore, the dose given per body weight must be an error that needs to be corrected in main text and Table 3. The XN and TXN compounds are formulated with the diet, therefore the daily dose of compound consumed by each mouse will vary depending on the quantity of food consumed and the body weight of the animal, which increases over the period of the study. It would be preferable to indicate the percentage of compound in the diet, which is 0.003% TXN (Table 3, 0.003 g /100 g diet) and 0.003% or 0.006% XN (Table 3, 0.003 g or 0.006 g/100 g diet). The dose of compound per mouse over time could be estimated from body weights and average daily food intake data in the phenome_feeding.csv file.

The Reviewer made a very good suggestion. We have changed the original mg/kg dose unit to a percentage unit in the main text as well as in the corresponding tables.

4. Line 97 and 99 – Hyphens are not necessary in certain contexts (e.g. "TXN-supplementation") and not being used consistently throughout manuscript, (e.g. "XN supplementation") therefore please consider removing all such instances.

We have fixed these instances by removing the - throughout the manuscript.

5. In Lines 112 – 114 below, it is not clear whether the decreased food intake being referred to is a comparison within the HXN group (i.e. compared to week 0) or compared to the LFD group at the indicated weeks. Please clarify."HXN-treated mice adapted better to the HFD, indicated by decreased food intake at week 1, 6-10, 13, and 16 (p < 0.05), and caloric intake (p = 0.01) (Figure 2C-D), resulting in less BW gain."

We have clarified which comparison we made, at line # 116-117.

6. Figure 3, Supplement 2: It is unclear why the r and p values reported for this figure in the manuscript (e.g. Lines 217 – 222) are different from what is reported in the figure, please see excerpt below. Were different correlation tests used?…namely, there was an inverse relationship between caloric intake and plasma TAG among LFD mice (Spearman, r = -0.60, p = 219 0.04; Figure 3—figure supplement 2 A1), which was lost on the HFD (Spearman, r = 0.12, p = 0.70; Figure 3—figure supplement 2 A2). TXN treatment restored the negative correlation between caloric intake and plasma TAG (Spearman r = -0.65, p = 0.04; Figure 3—figure supplement 2 A5).

Yes, the Reviewer is correct. The statistics shown in the figure are linear regressions and corresponding p values; whereas statistics reported in the main text are spearman correlations and corresponding p values.

7. Line 215, define triglyceride (TAG)

We defined triglyceride (TAG) previously at line 52.

8. Line 238, energy expenditure was indicated in Figure 3C not 4C"In contrast to energy expenditure (Figure 3C)…"

We have corrected this mistake.

9. Line 291: change three to "3-fold vs. 2.5-fold increase…

We have corrected this as suggested.

10. Lines 289 – 292 describe data in three panels, Figure 5A – C (not just Figure 5 A, C).

We have corrected this mistake.

11. In Figure 5, the middle column of data should be labeled B1 – B5 and last columns of data should be labeled C1 – C5. Also, abbreviations used are found in text but not in Figure 5 legend therefore please add following labeling to Figure 5 for convenience of reader:– after A. Subcutaneous fat, please add (sWAT)– after B. Epididymal fat, please add (eWAT)– after C. Mesenteric fat, please add (mWAT)

Thank you for pointing this out to us. The mislabel was a careless mistake. We have fixed the labels and put the abbreviations as suggested for convenience of reader.

12. For clarity, please specify p = 0.06 in Figure 5B. Legend indicates that p< 0.05 was considered significant, therefore please consider changing wording in Lines 293- 296, to indicate that TXN-treated mice trended higher than HFD group:"Compared to the HFD group, a smaller but significant increase in eWAT adipose tissue weight was observed in HXN-treated mice while that of TXN-treated mice trended higher (p = 0.06) (Figure 5B)."

Thank you for the suggestion. We have clarified accordingly.

13. Define abbreviation in Line 336: low density lipoprotein receptor knock-out (LDLR-/-)

We defined the abbreviation as suggested.

14. Consider restructuring the sentence below for simpler comprehension and indicate Figure 6A at end of sentence.Change lines 338-340: "Supplementation with XN decreased in a dose-dependent manner the number and size of intrahepatic lipid vacuoles in HFD mice"."XN supplementation decreased the number and size of intrahepatic lipid vacuoles in HFD mice in a dose-dependent manner (Figure 6A)."

We have restructured the sentence as suggested.

15. Line 341: At end of sentence please specify (Figure 6A).

We specified as suggested.

16. Line 347: At end of sentence please specify (Figure 6B).

We specified as suggested.

17. Please consider below the reorganized description of Figure 3B results (lines 353- 358) for better clarity:Both HXN and TXN supplementation decreased liver lipid accumulation on a HFD by two-fold (Figure 3B). Three of 12 HXN-supplemented mice and 5 of 11 TXN-supplemented mice had less than 5% lipid area while 7 of 12 HXN-supplemented mice 9 of 11 TXN-supplemented mice had less than 10% lipid area (Figure 3B). In comparison, only 1 of 12 HFD control mice were below 10% lipid area (Figure 3B).

We reorganized this paragraph as suggested.

18. Line 421 please define abbreviation: Furthermore, Kyoto Encyclopedia of Genes and Genomes (KEGG) pathway…

We defined abbreviation as suggested.

19. Line 458 please define abbreviation: We implemented support vector machine (SVM).

We defined abbreviation as suggested.

20. Missing the word "are" at beginning of Line 469: are known target genes of…

We added the missing word are as suggested.

21. Line 507: PPARγ has two major isoforms, γ1 and γ2, generated from the same gene by alternative splicing, so it is unclear why Pparγ2 is indicated as "a predicted PPAR target gene". There is no description of relevance of Plin4 gene, which is presented in Figure 10, and Figure 10 legend does not mention Plin4. Please address.

Thank you for pointing this out. We have corrected this and added the Plin4 description as suggested. Line 649, Consistent with RNAseq results, TXN-treated mice had significantly lower expression of PPARγ and major PPARγ target genes *Cidec*, *Mogat1*, *Plin4* and *PPARγ2*.

22. Results section for Figures 11 and 12 seem to be scrambled. To improve clarity, please consider making the following rearrangements and edits to lines 537 – 547:"we tested escalating doses of XN and TXN (Strathmann and Gerhauser, 2012). After treatments, we determined the number of live cells using an MTT assay. XN and TXN were only significantly cytotoxic for 3T3-L1 cells at a dose of 50 μM (data not shown). While it is difficult to translate in vivo doses to in vitro doses, based on previous in vitro studies (Yang et al., 2007; Samuels, Shashidharamurthy and Rayalam, 2018) and our current cell viability data, we selected low (5 μM), medium (10 μM) and high (25 μM) concentrations of XN and TXN for the subsequent experiments where cell viability was greater than 90% (data not shown). Murine preadipocyte 3T3-L1 differentiation and …"Also, the sentence shown below, presently Lines 538 – 540 seems to be information pertaining to Figure 12 and should be moved to section 8:3T3-L1 cells were treated with 0.1% DMSO, 1 μM rosiglitazone (RGZ), 1 μM GW9662, XN (5, 10 and 25 μM), TXN (5, 10 and 25 μM), 25 μM XN + 1 μM RGZ or 25 μM TXN + 1 μM RGZ for 48 h.

Thank you for pointing this out. We have rearranged the corresponding texts as suggested.